# Phosphorylation of plasma membrane H⁺-ATPase Thr881 participates in light-induced stomatal opening

Yuki Hayashi[1,3], Kohei Fukatsu[1,3], Koji Takahashi [1,2,3], Satoru N. Kinoshita [1], Kyohei Kato[1], Taku Sakakibara[1], Keiko Kuwata[2] & Toshinori Kinoshita [1,2] ✉

Plasma membrane (PM) H⁺-ATPase is crucial for light-induced stomatal opening and phosphorylation of a penultimate residue, Thr948 (pen-Thr, numbering according to *Arabidopsis* AHA1) is required for enzyme activation. In this study, a comprehensive phosphoproteomic analysis using guard cell protoplasts from *Vicia faba* shows that both red and blue light increase the phosphorylation of Thr881, of PM H⁺-ATPase. Light-induced stomatal opening and the blue light-induced increase in stomatal conductance are reduced in transgenic *Arabidopsis* plants expressing mutant AHA1-T881A in *aha1−9*, whereas the blue light-induced phosphorylation of pen-Thr is unaffected. Auxin and photosynthetically active radiation induce the phosphorylation of both Thr881 and pen-Thr in etiolated seedlings and leaves, respectively. The dephosphorylation of phosphorylated Thr881 and pen-Thr are mediated by type 2 C protein phosphatase clade D isoforms. Taken together, Thr881 phosphorylation, in addition of the pen-Thr phosphorylation, are important for PM H⁺-ATPase function during physiological responses, such as light-induced stomatal opening in *Arabidopsis thaliana*.

Stomata surrounded by a pair of guard cells in the plant epidermis control gas exchange between leaves and the atmosphere in response to environmental stimuli such as light, humidity, and $CO_2$, thereby regulating the balance between desiccation and photosynthesis[1,2]. Stomatal opening, which enhances $CO_2$ uptake from the atmosphere for photosynthesis, transpiration, and flow-mediated nutrient accumulation from soil, has a profound effect on plant growth and yields[3–5]. Light induces stomatal opening through a guard-cell-autonomous mechanism and through intercellular signal transduction. Blue light is perceived by phototropins in guard cells, such that the signal is transduced to several protein kinases (BLUE LIGHT SIGNALING1; BLUS1 and BLUE LIGHT-DEPENDENT H⁺-ATPASE PHOSPHORYLATION; BHP) and protein phosphatase (PP1), resulting in phosphorylation of a penultimate residue, a Thr948 (pen-Thr, numbering according to *Arabidopsis thaliana* H⁺-ATPase 1, AHA1), in the C-terminus of plasma membrane (PM) H⁺-ATPase[6–10]. The ensuing interaction between

14–3–3 protein and the phosphorylated pen-Thr activates PM H⁺-ATPase, which in turn promotes PM hyperpolarization and drives K⁺ uptake via voltage-gated inward-rectifying K⁺ channels. The accompanying uptake of water causes the flanking guard cells swell, and their arcing away from each other enlarges the stomatal aperture[1,2]. Red light also induces stomatal opening in intact leaves, again by enhancing the phosphorylation of pen-Thr in guard-cell PM H⁺-ATPase, most likely via photosynthesis in mesophyll cells and intercellular signal transmission[11,12]. Genetic analyses using loss-of-function mutants of a major PM H⁺-ATPase isoform in guard cells have clearly shown that the enzyme is essential for light-induced stomatal opening[11,13–15]. PM H⁺-ATPase is a P-type ATPase that actively transports H⁺ from the inside to the outside of the plant cells using energy from ATP hydrolysis[16]. It is expressed in all plant cell types examined so far. The transport of H⁺ results in the formation of an electrochemical potential gradient across the plasma membrane that is coupled with secondary

[1]Graduate School of Science, Nagoya University, Chikusa, Nagoya, Japan. [2]Institute of Transformative Bio-Molecules (WPI-ITbM), Nagoya University, Chikusa, Nagoya, Japan. [3]These authors contributed equally: Yuki Hayashi, Kohei Fukatsu, Koji Takahashi. ✉e-mail: kinoshita@bio.nagoya-u.ac.jp

transporters of minerals, sugars, and amino acids. These compounds are needed for physiological processes such as cell elongation, phloem loading, nutrient uptake in roots, and stomatal opening. An increase in the phosphorylation of pen-Thr of PM H+-ATPase in mesophyll cells and hypocotyl cells, by light and plant hormones such as auxin and brassinosteroid, respectively, also induces nutrient uptake and hypocotyl cell elongation[17–21].

Post-translational modifications of PM H+-ATPase occur not only at pen-Thr but also at several other phosphorylation sites, including Thr881, Ser899, and Ser931 (numbering according to *Arabidopsis* AHA1)[2,22–24]. Among these, several reports suggest that phosphorylation of Thr881 may be important for PM H+-ATPase activity. In yeast expressing plant PM H+-ATPase in which Thr881 was replaced with Ala, the growth rate was reduced whereas the replacement of Thr881 with Asp enhanced plant growth under acidic conditions[25–27]. These results demonstrate the positive effect of Thr881 phosphorylation on PM H+-ATPase activity. Nonetheless, whether Thr881 phosphorylation is involved in the regulation of PM H+-ATPase in guard cells and in stomatal opening in response to light is unknown.

Recently, Wong et al.[28] reported that type 2 C protein phosphatase clade D isoforms (PP2C.Ds) control stomatal movement. Akiyama et al.[29] showed that PP2C.Ds directly dephosphorylate phosphorylated pen-Thr of PM H+-ATPase in guard cells. More recently, an increase in the intercellular $CO_2$ concentration in leaves, by the termination of light irradiation, induced the rapid dephosphorylation of phosphorylated pen-Thr in guard cells, mediated by PP2C.Ds, and stomatal closing[12]. However, the protein phosphatases that direct the dephosphorylation of phosphorylated Thr881 have yet to be identified. Regarding the protein kinase for Thr881, the receptor kinase PLANT PEPTIDE CONTAINING SULFATED TYROSINE 1 RECEPTOR (PSY1R), as well as BRI1-associated receptor kinase (BAK1), have been proposed as candidate protein kinases responsible for Thr881 phosphorylation[27,30]. Note that Transmembrane kinase 1 (TMK1) and TMK4 have been shown to mediate pen-Thr phosphorylation in seedlings and roots[31,32].

This study consisted of a comprehensive phosphoproteomic analysis using guard-cell protoplasts (GCPs) from *Vicia faba*, a common plant from which large numbers of GCPs can be isolated with high purity[6,33]. Phosphorylated Thr881, Ser899, Thr942, and Thr948 (pen-Thr) (numbering according to *Arabidopsis* AHA1) were confirmed in *Vicia faba* PM H-ATPase isoforms (VHAs), and Thr881 was phosphorylated in response to red and blue light. Furthermore, red- and blue-light-induced phosphorylation of Thr881 were also observed in guard cells from *Arabidopsis thaliana*. In transgenic *Arabidopsis* plants

in which Thr881 of the PM H+-ATPase was replaced by Ala, both light-induced stomatal opening in the epidermis and the light-induced increase in transpiration in intact leaves were reduced. The phosphorylation of Thr881 in other tissues and cell types in response to physiological signals and the involvement of PP2C.Ds in the dephosphorylation of phosphorylated Thr881 were determined as well. These results are consistent with a role for PM H+-ATPase Thr881 phosphorylation in physiological responses, such as light-induced stomatal opening in *Arabidopsis thaliana*.

## Results

### Multiple phosphorylation of PM H+-ATPase in response to red and blue light in stomatal guard cells

PM H+-ATPase is phosphorylated at several sites, including Thr881, Ser899, Ser931, and Thr948 (numbering according to *Arabidopsis* AHA1), all of which are located at the C-terminal autoinhibitory stretch of the enzyme[2,22–24]. Thr948 is the penultimate residue (pen-Thr), and its phosphorylation status regulates PM H+-ATPase activity. In guard cells, the phosphorylation status of pen-Thr in PM H+-ATPase has been intensively investigated[34,35], whereas the other phosphorylation sites of PM H+-ATPase in guard cells have not been well studied. In this study, a comprehensive phosphoproteomic analysis was performed using GCPs from *Vicia faba*. This species was chosen because GCPs can be obtained at higher purity and in larger amounts (>1 mg protein) than from *Arabidopsis thaliana*. However, because genomic information for *Vicia faba* had been lacking at start of this study, its proteomic analysis in this work was achieved by constructing an expression database from the plant[36]. Note that, later, the genome of *Vicia faba* has been sequenced[37].

In *Vicia faba* GCPs, blue light-induced phosphorylation of phototropin orthologs, vfphot1a and vfphot1b, and pen-Thr of PM H+-ATPases[6,33]. In addition, we also found that an ortholog of BLUS1, a blue light-signaling component identified in *Arabidopsis*[9], was phosphorylated in response to blue light (Supplementary Table 1). These results indicate that *Vicia* GCPs using this study have ability to respond to blue light signal. Interestingly, we found that the phosphorylation level of Thr881 of PM H+-ATPase isoforms, including VHA1 and an unnamed PM H+-ATPase isoform, increased in response to both red and blue light, although pen-Thr (Thr948) in *Vicia* GCPs was phosphorylated mainly in response to blue light (Table 1, Supplementary Table 2). Furthermore, we found that blue light-induced phosphorylation of Thr942. Ser899 was phosphorylated but there was no change in its phosphorylation status in response to light. The fungal toxin fusicoccin (FC), an activator of PM H+-ATPase, increased the phosphorylation of both pen-Thr and Thr881 in *Vicia* GCPs (Supplementary Table 3).

It is worthy of note that we found several additional blue light-induced phosphorylated proteins, such as LMBR1-like membrane protein and ARM repeat superfamily protein, in *Vicia* GCPs (Supplementary Table 1). Further analysis will be needed to investigate the properties and functions in the blue light-signaling pathway of these candidate proteins.

Next, we confirmed the phosphorylation behavior of Thr881 and pen-Thr in guard cells from *Arabidopsis thaliana* using the respective specific antibodies (See Methods, Supplementary Fig. 1). Figure 1a shows the results of immunohistochemical analysis of phosphorylated Thr881 and pen-Thr in guard cells of isolated epidermal tissues. Red light irradiation (R) induced Thr881 phosphorylation but not pen-Thr phosphorylation. However, blue light irradiation under background red light (RB) enhanced the phosphorylation of both Thr881 and pen-Thr. The level of Thr881 phosphorylation induced by simultaneous blue and red light irradiation (RB) was significantly higher than that induced by red light irradiation (R alone), indicating that blue light drives both Thr881 and pen-Thr phosphorylation. FC-induced pen-Thr phosphorylation but not Thr881 phosphorylation in guard cells from *Arabidopsis* epidermis without changing PM H+-ATPase amount

**Table 1 | Phosphoproteome analysis of *Vicia* guard-cell protoplasts (GCPs) illuminated with red and blue light**

| Phospho-site | Contig | PSMs | | | Fold change | |
|---|---|---|---|---|---|---|
| | | Dk | R | R + B | ([R]/[Dk]) | ([R + B]/[R]) |
| Thr881 | VHA1 | 11 | 26 | 65 | 2.36 | 2.50 |
| | Unnamed isoform | 0 | 1 | 10 | - | 10.00 |
| Ser899 | VHA1 | 12 | 19 | 14 | 1.58 | 0.74 |
| | Unnamed isoform | 6 | 10 | 9 | 1.67 | 0.90 |
| Thr942 | VHA1/Unnamed isoform | 8 | 9 | 23 | 1.13 | 2.56 |
| Thr948 (pen-Thr) | VHA1/Unnamed isoform | 50 | 49 | 89 | 0.98 | 1.82 |

GCPs were isolated from *Vicia faba* and kept in the dark for 1 hr. Dark-acclimated GCPs were kept in the dark for more 30 min (Dk) or illuminated with red light (600 µmol m⁻² s⁻¹) for 30 min (R), after which blue light (100 µmol m⁻² s⁻¹) was simultaneously illuminated with the red light for 1 min (R + B). Peptide spectrum matches (PSMs), the total number of identified peptide spectra matched for the protein, of each phosphopeptides from VHA1 and unnamed isoform and fold change are indicated (-; infinity). The numbers in the Phospho-site correspond to the amino-acid number of AHA1. Experiments were repeated on three occasions with similar results (Supplementary Table 2).

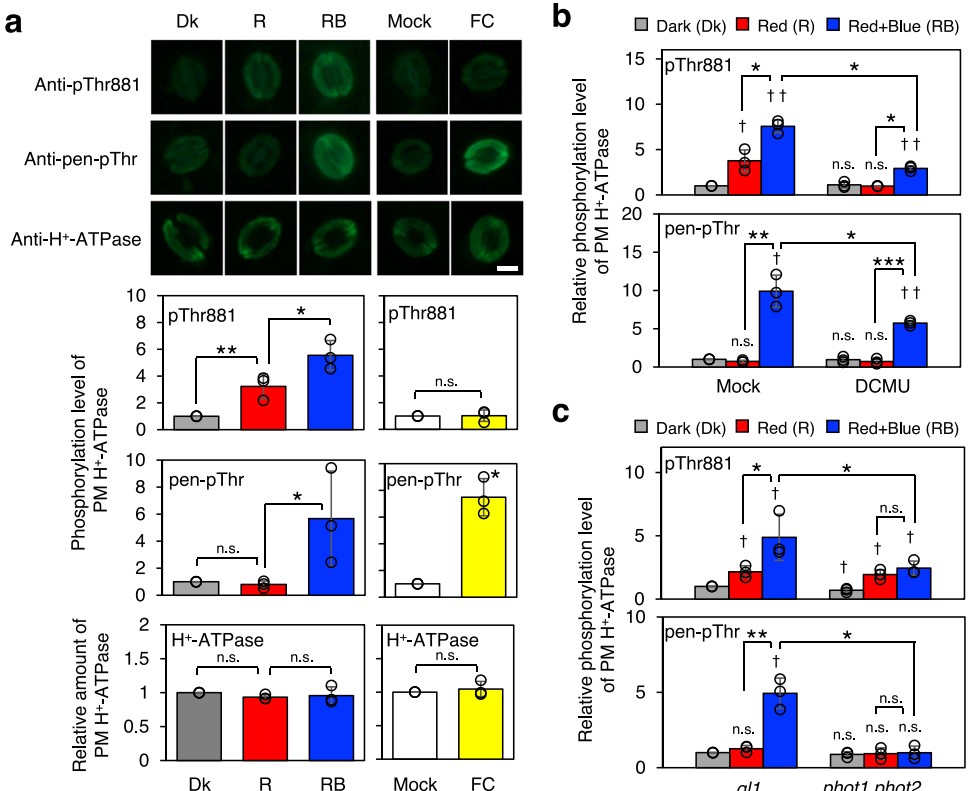

**Fig. 1 | Immunohistochemical estimation of the phosphorylation levels in the guard cells of Arabidopsis PM H⁺-ATPase. a** Detection of PM H⁺-ATPase phosphorylation in Arabidopsis guard cells. Leaf epidermal tissues isolated from dark-acclimated *Arabidopsis thaliana* Col-0 plants were incubated in the basal buffer and kept in the dark for 30 min (Dk) or illuminated with red light and blue light. Red light (50 μmol m⁻² s⁻¹) was illuminated for 30 min (R), after which blue light (10 μmol m⁻² s⁻¹) was simultaneously illuminated with the red light for 2.5 min (R + B). The fungal toxin fusicoccin (FC) at 10 μM or DMSO (solvent control; Mock) was added to the basal buffer, and the epidermal tissues were incubated in the dark for 5 min. The phosphorylated PM H⁺-ATPase and PM H⁺-ATPase proteins were detected using anti-pThr881, anti-pen-pThr, and anti-H⁺-ATPase antibodies. Typical immunofluorescent images (top) and relative fluorescence intensities of fluorescent signals (bottom) are shown. Data represent averages of relative values from three independent experiments with standard deviation (SD). The asterisk indicates statistically significant difference (one-tailed Student's *t* test: *, ** and n.s [not significant] indicate $P < 0.05$, $P < 0.01$ and $P > 0.05$, respectively). Bar = 10 μm.

**b** Inhibition of red light-induced phosphorylation of Thr881 by photosynthesis inhibitor DCMU. Epidermal tissues were pretreated with 10 μM DCMU or EtOH (solvent control; Mock) for 20 min in the dark before light illumination. Daggers indicate that the phosphorylation levels are significantly higher than Dk of Mock set to 1 (two-tailed Student's *t* test: †, †† and n.s. indicate $P < 0.05$, $P < 0.01$, and $P > 0.05$). Asterisks indicate that the mean of RB are significantly higher than that of R within each experiment and that the averages of RB of DCMU are significantly lower than that of Mock (one-tailed Student's *t* test: ***, $P < 0.0001$). Others are same as in **a**. **c** Suppression of the blue light-induced Thr881 and pen-Thr phosphorylation in a *phot1 phot2* double mutant. Immunohistochemical methods were performed using *phot1 phot2* and *gl1*, the background plant. Daggers indicate that the phosphorylation levels are significantly higher than Dk of *gl1* set to 1 (two-tailed Student's *t* test: †, ††, n.s. indicate $P < 0.05$, $P < 0.01$, and $P > 0.05$, respectively). Asterisks indicate that the averages of RB are significantly higher than that of R within each genotype and that the averages of RB of *phot1 phot2* are significantly lower than that of *gl1* (one-tailed Student's *t* test: * indicates $P < 0.05$). Others are same as in **a**.

(Fig. 1a). Essentially, we obtained similar results in GCPs from *Arabidopsis thaliana* (Supplementary Fig. 2). These results indicate that blue and red light-induced Thr881 phosphorylation of PM H⁺-ATPase in guard cells are common at least both in *Vicia faba* and *Arabidopsis thaliana*. It is worthy of note that FC induces phosphorylation of Thr881 in *Vicia* GCPs (Supplementary Table 3), but not in guard cells from *Arabidopsis thaliana* (Fig. 1a and Supplementary Fig. 2).

Next the phosphorylation properties of Thr881 and pen-Thr in *Arabidopsis* guard cells were investigated via immunohistochemical analysis (Fig. 1b, c). Suppression of the red light-induced phosphorylation of Thr881 in *Arabidopsis* Columbia-0 (Col-0) by an inhibitor of photosynthetic electron transport, 3-(3,4-dichlorophenyl)−1,1-dimethylurea (DCMU), suggests that the reaction is mediated by photosynthesis in guard cells. Suppression of blue light-induced Thr881 and pen-Thr phosphorylation in a *phot1 phot2* double mutant indicates that the blue light receptors (phototropins) mediate blue light-induced Thr881 and pen-Thr phosphorylation. Then the phosphorylation status of Thr881 and pen-Thr in GCPs from a *bhp* mutant was determined (Fig. 2). BHP is a blue light-signaling component that interacts with

BLUS1 in guard cells[10]. In Col-0, the blue light-induced phosphorylation of pen-Thr and Thr881 was observed. Interestingly, phosphorylation level of Thr881 was constitutively high, although, blue-light-induced phosphorylation of pen-Thr was suppressed in *bhp* mutant as in the previous report[10].

## Phenotypic analysis of transgenic plants expressing AHA1-T881A, T881D and T948A

The importance of Thr881 phosphorylation in guard cells was examined by replacing Thr881 with Ala in AHA1 and then analyzing the phenotypes of transgenic plants carrying *AHA1-T881A* in a knockout mutant of *AHA1, aha1−9* (g*AHA1-T881A*), driven by the *AHA1* native promoter (Fig. 3). The amount of PM H⁺-ATPase was significantly decreased in the leaves and guard cells from a knockout mutant of *AHA1, aha1−9*, whereas in *AHA1-T881A* plants it was similar to that of wild-type plants (Fig. 3a, b). A knockout mutant of *AHA1, aha1−9*, showed reduced light-induced stomatal opening (Fig. 3c and Supplementary Fig. 3a, b) whereas in transgenic plants carrying wild-type *AHA1* in *aha1−9* (g*AHA1-WT*) light-induced stomatal opening was the

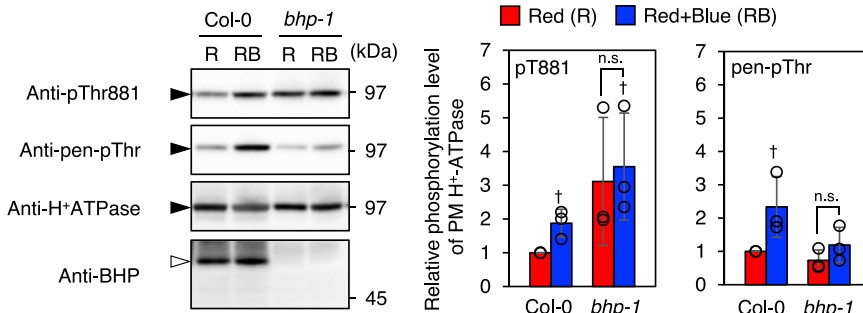

**Fig. 2 | Blue light-induced PM H⁺-ATPase phosphorylation in Col-0 and *bhp-1* GCPs.** GCPs isolated from *Arabidopsis thaliana* Col-0 and *bhp-1* were kept in the dark for 1 hr. Dark-adapted GCPs were illuminated by red light (R: 50 µmol m⁻² s⁻¹, 20 min), then blue light (RB: 10 µmol m⁻² s⁻¹, 2.5 min) was superimposed on the red light. The phosphorylated PM H⁺-ATPase and amount of PM H⁺-ATPase were detected by the immunoblot using anti-pThr881, anti-pen-pThr and anti-H⁺-ATPase antibodies. Black and white arrowheads show the positions of AHAs and BHP, respectively. The graphs on the right side show the relative phosphorylation level of PM H⁺-ATPase, which is calculated by dividing the signal intensities of the bands detected with anti-pThr881 or anti-pen-pThr by that of the band detected with anti-H⁺-ATPase antibody. The relative amount of PM H⁺-ATPase was calculated by normalizing against the amount of PM H⁺-ATPase in Red of Col-0. Data are mean ± SD of three independent experiments. Daggers indicate that the mean is statistically significantly higher than R of Col-0 (two-tailed Student's *t* test: †*P* < 0.05). n.s., Not significant (two-tailed Student's *t* test: *P* > 0.05).

same as in wild-type Col-0 (Supplementary Fig. 3). By contrast, light-induced stomatal opening in *gAHA1-T881A* plants was significantly reduced compared to Col-0 plants (Fig. 3c). Consistent with these results, the blue light-induced increase in transpiration was also significantly reduced in *gAHA1-T881A* (Fig. 3d, e). These results indicate an important role for Thr881 phosphorylation in PM H⁺-ATPase activity in guard cells of *Arabidopsis thaliana*.

The blue light-induced phosphorylation of pen-Thr in *gAHA1-T881A* plants was also investigated immunohistochemically. The replacement of Thr881 with Ala had no effect on the blue light-induced phosphorylation of pen-Thr (Fig. 3f). In a previous study, the T881A mutation had no effect on the binding of 14–3–3 protein to the C-terminus, including phosphorylated pen-Thr, of PM H⁺-ATPase expressed in yeast[27]. The ability of the PM H⁺-ATPase activator FC to induce stomatal opening in *gAHA1-T881A* plants (Fig. 3g) suggested that replacement of Thr881 by Ala does not induce the conformational change in PM H⁺-ATPase which affects the catalytic activity. We further investigated the phenotypes in Thr881 phospho-mimic transgenic plants having *AHA1-T881D* in *aha1-9* (*gAHA1-T881D*) (Supplementary Fig. 4). *gAHA1-T881D* plants showed slightly higher average of light-induced stomatal opening with showing normal blue light-induced phosphorylation of pen-Thr. These results suggest that the phosphorylation of Thr881 affects PM H⁺-ATPase activity but has no effect on the phosphorylation status of pen-Thr, and that, in the presence of phosphorylated pen-Thr, Thr881 phosphorylation additively enhances PM H⁺-ATPase activity.

In addition, we investigated light-induced stomatal opening in transgenic plants having *AHA1-T948A* in *aha1-9* (*gAHA1-T948A*) (Fig. 3h, i). Light-induced stomatal opening was completely suppressed in *gAHA1-T948A* plants, indicating that phosphorylation of pen-Thr in PM H⁺-ATPase is critical for the enzyme's activity in guard cells.

### Phosphorylation of Thr881 in leaves and etiolated seedlings in response to physiological stimulus

It has been demonstrated that photosynthetic active radiation (PAR) induces the phosphorylation of pen-Thr in PM H⁺-ATPase via photosynthesis in photosynthetic tissues from not only vascular plants but also non-vascular plants[18,19,21]. In addition, the plant hormone auxin enhances pen-Thr phosphorylation during the auxin-induced elongation growth of hypocotyls[17,31,32,38]. As shown in Figs. 1–3, in guard cells, blue light induced both phosphorylations of pen-Thr and Thr881. However, so far, phosphorylation status of Thr881 in photosynthetic tissues and hypocotyls has not been investigated. So, we first investigated the phosphorylation behavior of Thr881 of PM H⁺-ATPase in

mesophyll cells by an immunoblot analysis of *Arabidopsis* leaves irradiated with PAR for 30 min. PAR significantly increased the phosphorylation of Thr881 and pen-Thr in PM H⁺-ATPases in leaves (Fig. 4a). Next, we investigated effect of 10 µM auxin indole-3-acetic acid (IAA) on phosphorylation status of Thr881 in hypocotyl segments excised from etiolated *Arabidopsis* hypocotyls. The results showed that both Thr881 and pen-Thr phosphorylation increased, as determined by immunoblot analysis (Fig. 4b). These results demonstrate that Thr881 phosphorylation occurs in response to diverse environmental signals that enhance the activity of PM H⁺-ATPase. Furthermore, the signaling pathway responsible for regulating Thr881 phosphorylation was found to function not only in guard cells but also in mesophyll cells and etiolated hypocotyls.

The fungal toxin FC, an activator of PM H⁺-ATPase, strongly reduced Thr881 phosphorylation whereas it increased pen-Thr phosphorylation in etiolated hypocotyl segments (Supplementary Fig. 5). Conversely red light irradiation induced only Thr881, not pen-Thr phosphorylation in guard cells, indicative of independent molecular mechanisms in the regulation of Thr881 and the pen-Thr phosphorylation. This suggestion was further supported in experiments conducted in mesophyll cell protoplasts (MCPs), in which the effects of the transient expression of GFP-AHA1-WT (wild type), -T881A, -T881D and -T948A were examined. In the MCPs, GFP-AHA1-WT was phosphorylated both at Thr881 and at pen-Thr. In GFP-AHA1-T881A and -T881D, pen-Thr was phosphorylated to the same extent as in GFP-AHA1-WT, while in GFP-AHA1-T948A Thr881 was phosphorylated to the same extent as in GFP-AHA1-WT (Supplementary Fig. 1c).

### PSY1R and its isoforms don't mediate phosphorylation of Thr881

A previous study showed that a recombinant PSY1R phosphorylates Thr881 in vitro, and that a *psy1r* null mutant displays lower phosphorylation level of Thr881 in planta. Furthermore, treatment of PSY1 peptide, a ligand for PSY1R, induced Thr881 phosphorylation, but not in the *psy1r* mutant[27]. These results strongly suggest that PSY1R mediate phosphorylation of Thr881. To confirm this, we investigated phosphorylation status of Thr881 in a *psy1r-1* single mutant and a triple mutant of *PSY1R* and two isoforms, *PSKR1* and *PSKR2*[39]. As shown in Fig. 4c, in Col-0 leaves, PAR (Lt) induced phosphorylation of Thr881 and pen-Thr. PAR also induced normal phosphorylation response on Thr881 and pen-Thr in not only *psy1r* single mutant, but also *psy1r pskr1 pskr2* triple mutant (*TKO*), suggesting that PSY1R and its isoforms may not mediate phosphorylation of Thr881 in *Arabidopsis* leaves. In support with this, another study also showed that treatments of PSY1 and

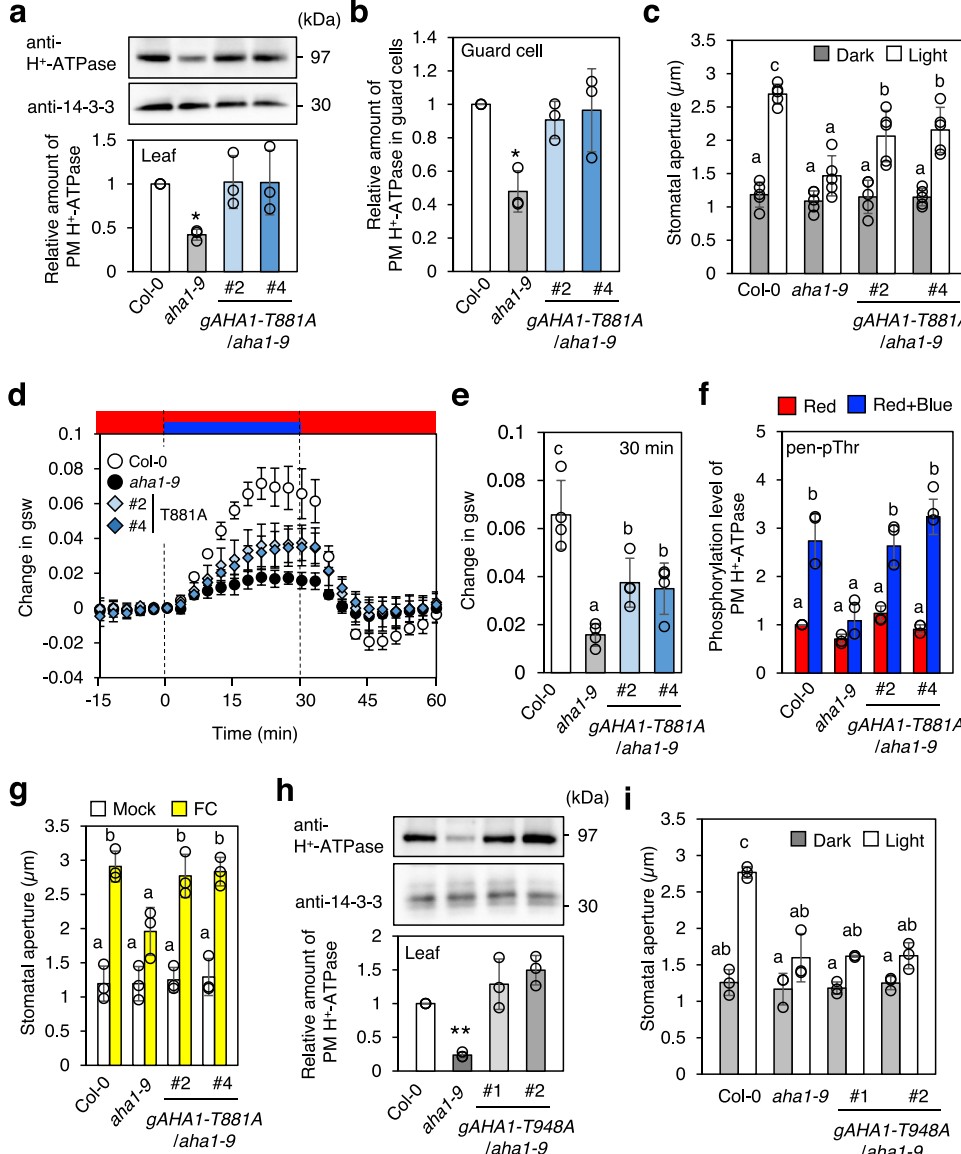

**Fig. 3 | Effect of phospho-defective form of Thr881 and pen-Thr (Thr948) in PM H⁺-ATPase on stomatal responses. a** Expression of Thr881 phospho-defective form (T881A) of AHA1 in *aha1-9* mutant (*gAHA1-T881A/aha1-9*). Immunoblots of PM H⁺- ATPase and 14-3-3 proteins were carried out using proteins obtained from rosette leaves of 4-week-old plants with specific antibodies. The 14-3-3 proteins were used as a loading control. Relative amounts of PM H⁺-ATPase were estimated from the total PM H⁺-ATPase signals normalized against that in Col-0. Data are mean ± SD of three independent experiments. Asterisk indicates a significant statistical difference relative to Col-0 (two-tailed Student's *t* test: *$P < 0.01$).
**b** Expression of PM H⁺-ATPase in guard cells of *gAHA1-T881A* plants. PM H⁺-ATPase in guard cells were detected by immunofluorescence method using the specific antibody. Relative amounts of PM H⁺-ATPase were normalized against that in Col-0. Data are mean ± SD of three independent experiments. Asterisk indicates a significant difference relative to Col-0 (two-tailed Student's *t* test: *$P < 0.01$) **c** Light-induced stomatal opening in *gAHA1-T881A* plants. Leaf epidermal tissues from dark-acclimated plants in the basal buffer were incubated for 4 h with or without light (blue light at 10 μmol m⁻² s⁻¹ superimposed on red light at 50 μmol m⁻² s⁻¹). Values represent means ± SD ($n = 5$, independent experiments); measurement of 30 stomata in each experiment. Different letters indicate significant differences among means (ANOVA with Tukey's test: $P < 0.05$). **d** Light-dependent changes in

stomatal conductance. Leaves from dark-acclimated plants were illuminated with red light (Red: 600 μmol m⁻² s⁻¹) and/or blue light (60 μmol m⁻² s⁻¹) as indicated. Data are mean ± SD of four independent experiments. **e** Stomatal conductance at 30 min after starting blue light illumination. Data were obtained from **d**. Different letters indicate significant differences among means (ANOVA with Tukey's test: $P < 0.05$). **f** Immunohistochemical detection of the pen-Thr phosphorylation in guard cells in response to blue light in *gAHA1-T881A* plants. Data are mean ± SD of three independent experiments. Different letters indicate statistically significant differences among means (ANOVA with Tukey's test: $P < 0.05$). Others are the same as in Fig. 1a. **g** Stomatal opening in response to FC. Epidermal tissues from dark-acclimated plants in the basal buffer were treated with DMSO (Mock) or 10 μM FC (FC) in the dark for 4 h. Data are mean ± SD of three independent experiments. Different letters indicate statistically significant differences among means (ANOVA with Tukey's test: $P < 0.05$). Other details are the same as in **c**. **h** Expression of Thr948 phospho-defective form (T948A) of AHA1 in *aha1-9* mutant (*gAHA1-T948A/aha1-9*). Asterisk indicates a significant statistical difference relative to Col-0 (two-tailed Student's *t* test: **$P < 0.01$). Other details are the same as in **a**. **i** Light-induced stomatal opening in *gAHA1-T948A* plants. Data are mean ± SD ($n = 3$, independent experiments); measurement of 30 stomata in each experiment. Different letters indicate significant differences among means (ANOVA with Tukey's test: $P < 0.05$).

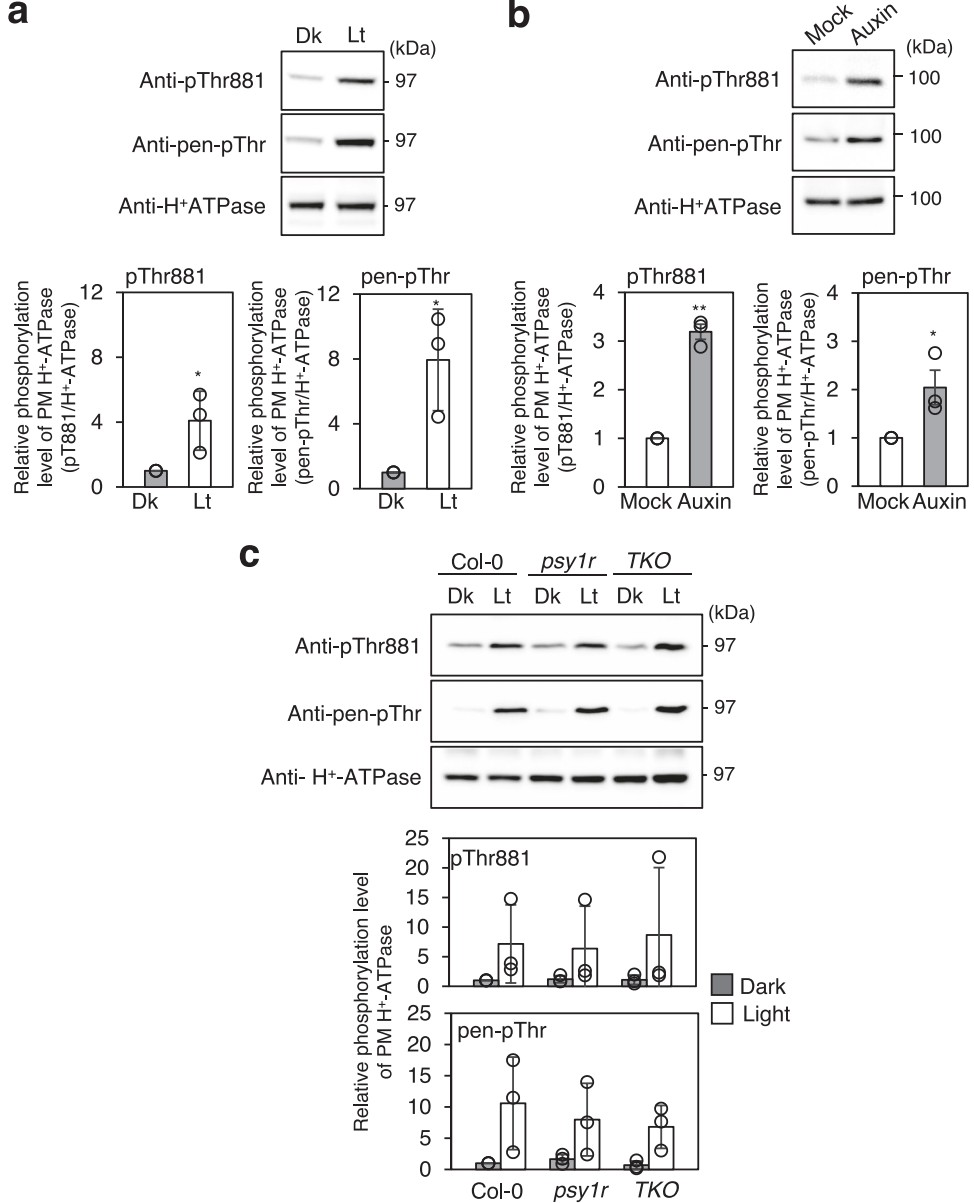

**Fig. 4 | Phosphorylation of PM H⁺-ATPases in etiolated hypocotyl segments and leaves. a** Light-induced phosphorylation of PM H⁺-ATPase in rosette leaves. Leaf pieces from dark-acclimated 4-week-old Arabidopsis plants were illuminated with white light ($50\ \mu mol\ m^{-2}\ s^{-1}$; Lt) or kept in the dark (Dk) for 30 min. The phosphorylation level and amount of PM H⁺-ATPase were estimated by the immunoblot using anti-pThr881, anti-pen-pThr and anti-H⁺-ATPase antibodies. Arrowheads show the position of AHAs. Lower graphs show the relative phosphorylation level of PM H⁺-ATPase, which is calculated by dividing the signal intensities of the bands detected with anti-pThr881 or anti-pen-pThr by that of the band detected with anti-H⁺-ATPase antibody. Data are mean ± SD of three independent experiments. The asterisks indicate significant difference compared to DMSO (one-tailed Student's $t$ test: *$P < 0.05$). **b** Auxin-induced phosphorylation of PM H⁺-ATPase in etiolated hypocotyl segments. Hypocotyl segments excised from 3-day-old Arabidopsis etiolated seedlings were treated with DMSO (Mock) or $10\ \mu M$ IAA (Auxin) in the dark for 30 min. Data are mean ± SD of three independent experiments. The asterisk indicates a significant difference (two-tailed Student's $t$ test: *$P < 0.05$; **$P < 0.0001$). Others are the same as in **a**. **c** Leaf pieces from dark-adapted Col-0, *psy1r* and *psy1r pskr1 pskr2* triple mutant (*TKO*) were illuminated with white light. Others are same as in **a**.

PSY5 peptides with *Arabidopsis* seedlings have no effect on phosphorylation status of Thr881[40].

## Dephosphorylation of phosphorylated Thr881 by PP2C.Ds

A previous study has demonstrated that PP2C.D isoforms mediate the direct dephosphorylation of phosphorylated pen-Thr of PM H⁺-ATPase in guard cells[29]. Thus, the protein phosphatase mediating the dephosphorylation of phosphorylated Thr881 was investigated. An in vitro dephosphorylation assay using microsomes from *Arabidopsis* leaves (Fig. 5a) showed a decrease in Thr881 and pen-Thr phosphorylation in microsomes after 30 min in vitro incubation. The inhibition of this

effect by adding EDTA implicated a divalent-cation-dependent protein phosphatase, such as PP2C, in dephosphorylation. While FC induces the tight binding of 14–3–3 protein to suppresses the dephosphorylation of phosphorylated pen-Thr[34,41], it did not suppresses the dephosphorylation of phosphorylated Thr881 (FC), ruling out a role for 14–3–3 protein in Thr881 dephosphorylation.

Next, the ability of PP2C.D to dephosphorylate phosphorylated Thr881 in vivo was investigated in transient PP2C expression assays using *Arabidopsis* MCPs (Fig. 5b, Supplementary Fig. 6). Consistent with previous results[29], the transient expression of GFP-PP2C.D6 but not GFP and GFP-PP2C.C4 induced the dephosphorylation of

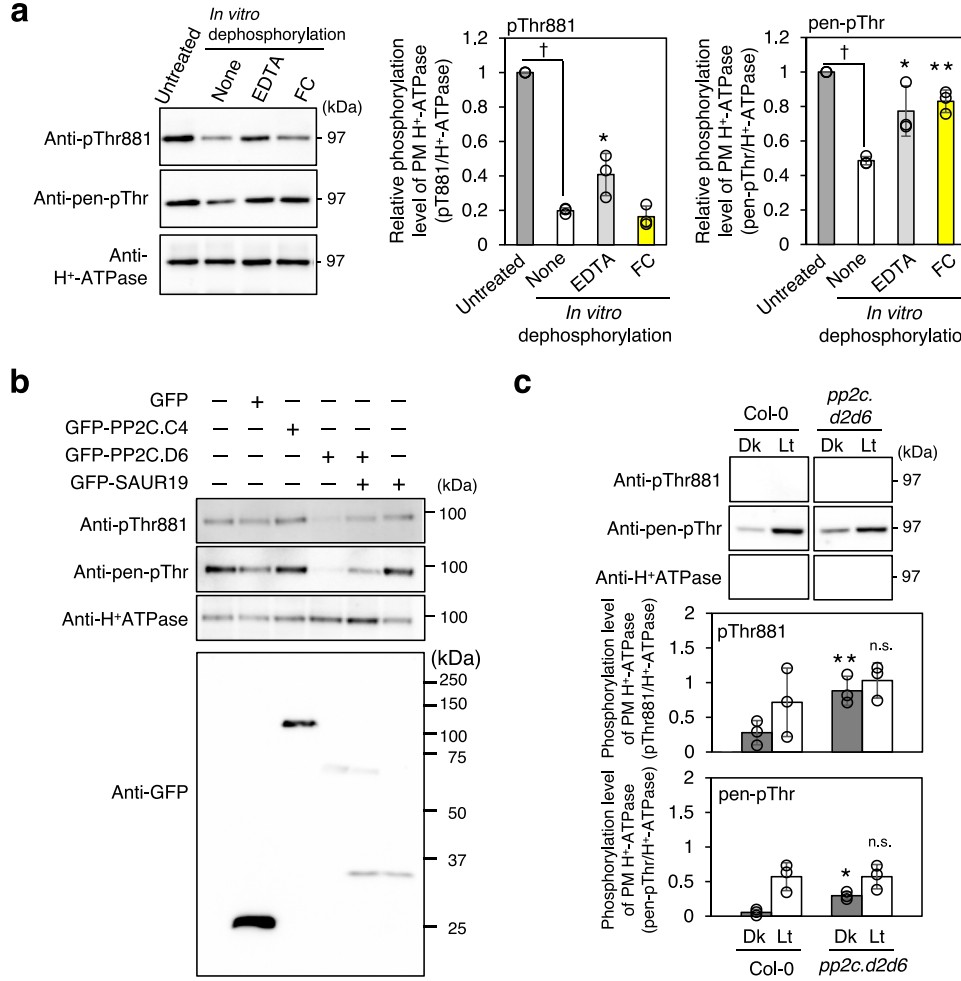

**Fig. 5 | Dephosphorylation of Thr881 in PM H⁺-ATPases by type-2C protein phosphatase (PP2C). a** In vitro dephosphorylation of PM H⁺-ATPase in the microsomes isolated from leaves. Microsomal membranes isolated from Col-0 leaves were used for the dephosphorylation (Untreated) and the reactions were carried out for 30 min at 24 °C (None). EDTA at 0.5 mM and 10 μM FC were added to the reaction buffer. Daggers indicate that the mean is significantly higher than Untreated set to 1 (two-tailed Student's $t$ test: †$P < 0.0001$). Asterisk indicates a significant difference relative to None (Dunnett's test: *$P < 0.05$; **$P < 0.01$). Others are same as in Fig. 2. **b** Transient expression of *GFP-PP2Cs* and *GFP-SAUR19* in the mesophyll cell protoplasts. GFP-PP2Cs and GFP-SAUR19 were detected using anti-GFP antibody. Numbers at right indicate molecular weight markers. Others are

same as in Fig. 2. Experiments were repeated on three occasions with similar results (Supplementary Fig. 6). **c** Light-induced phosphorylation of PM H⁺-ATPase in *pp2c.d* mutants. Leaf pieces from dark-adapted Col-0 and *pp2c.d2d6* were illuminated with white light (Lt) or kept in the dark (Dk) for 30 min. Lower graphs show the phosphorylation level of PM H⁺-ATPase, which is calculated by dividing the signal intensities of the bands detected with anti-pThr881 or anti-pen-pThr by that of the band detected with anti-H⁺-ATPase antibody. Data are mean ± SD of three independent experiments. The asterisks indicate a statistically significant difference between Col-0 and *pp2c.d2d6* (one-tailed Student's $t$ test: *$P < 0.05$; **$P < 0.01$; n.s., Not significant, $P > 0.05$). Others are the same as in Fig. 4a.

phosphorylated pen-Thr. Moreover, phosphorylated Thr881 was also dephosphorylated by PP2C.D6. Both dephosphorylation reactions were suppressed by the simultaneous expression of the PP2C.D inhibitor SAUR with PP2C.D6.

Finally, we investigated phosphorylation status of Thr881 of PM H⁺-ATPase in leaves from Col-0 and *pp2c.d* mutant (Fig. 5c). We used *pp2c.d2d6* double mutant because these isoforms show relatively higher expression level in leaves (Supplementary Fig. 7). The results showed that *pp2c.d2d6* showed higher phosphorylation levels of Thr881 and pen-Thr than those in Col-0 in the dark. These results support a role for PP2C.D isoforms in the dephosphorylation of phosphorylated pen-Thr and phosphorylated Thr881 in plant cells.

## Discussion

This study reveals that Thr881 is phosphorylated by red and blue light in stomatal guard cells, and Thr881 phosphorylation affects PM H⁺-ATPase function. Previously, Thr881 phosphorylation and its effect on PM H⁺-ATPase activity have been analyzed in yeast and in

comprehensive phosphoproteomic analyses in plant cells[25–27,30]. However, this is the first study to identify the physiological signals that induce Thr881 phosphorylation in plant cells and the role of Thr881 phosphorylation in the physiological responses mediated by PM H⁺-ATPase, such as stomatal opening. At the same time, in this study, we provide definitive results to show the importance of pen-Thr phosphorylation for PM H⁺-ATPase function.

Our investigations benefited from a combined approach using materials suitable for physiological and biochemical investigations and those more appropriate for molecular biology analyses. Comprehensive phosphoproteomics were performed using GCPs from *Vicia faba*, which are suitable for biochemical experiments, while molecular biological analyses were carried out in *Arabidopsis thaliana*, as the results could be linked to those of functional studies.

It is clear that Thr881 is phosphorylated by red and blue light. Although Thr881 phosphorylation definitely plays a positive role in stomatal opening, but is not as critical as the phosphorylation of pen-Thr (Fig. 3). In the isolated epidermis and GCPs from *Arabidopsis*

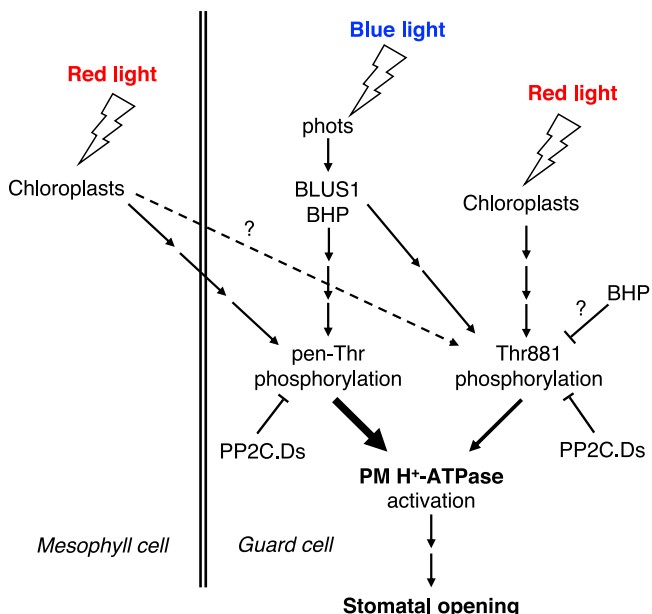

**Fig. 6 | Schematic model of light-induced stomatal opening and contribution of Thr881 phosphorylation of PM H⁺-ATPase in guard cells.** Arrows and T-bars lines denote positive and negative regulation, respectively. The thickness of the arrows connecting "phosphorylation" and "PM H⁺-ATPase activation" reflects their contribution.

*thaliana*, Thr881 phosphorylation in the guard cells was induced by both red and blue light, whereas pen-Thr phosphorylation was induced only by blue light (Fig. 1a). As shown in Fig. 1b, c in guard cells, the blue light-induced phosphorylation of Thr881 is mediated by phototorpins and the red light-induced phosphorylation of Thr881 by photosynthesis. Further investigations are needed to determine the mechanism by which red light, i.e., signals from photosynthesis via chloroplasts in guard cells, and phototropin downstream components induce Thr881 phosphorylation.

Why a knockout mutant of the blue light-signaling component BHP, *bhp-1*, resulted in constitutively high-level phosphorylation of Thr881 in guard cells (Fig. 2) is unclear, but it was not due to a defect of blue light signaling, because phosphorylation level of Thr881 in the *phot1 phot2* double mutant was not high unlike *bhp* mutant (Fig. 1c). The result also suggests that BHP is not required to phosphorylate Thr881 of the PM H⁺-ATPase in guard cells. Further studies may lead to insights into the role of BHP in guard-cell signal transduction.

Our study also showed that PP2C.D isoforms are involved in the dephosphorylation of phosphorylated Thr881 and pen-Thr (Fig. 5). However, why Thr881 is phosphorylated by both red and blue light, while pen-Thr is phosphorylated only by blue light, as well as how PP2C.D isoforms respond to red and blue light are thus far unknown. Then, what are the kinases involved in the phosphorylation of Thr881? Previous studies have suggested that PSY1R and BAK1 are involved in the phosphorylation of Thr881[27,30], but our results seem to rule out a role for PSY1R in the phosphorylation of Thr881 (Fig. 4c), as also suggested by another study[40]. Both TMK1 and TMK4 have been implicated in the phosphorylation of pen-Thr[31,32]. Whether these kinases are also involved in the phosphorylation of Thr881 awaits further investigations.

How does Thr881 phosphorylation lead to PM H⁺-ATPase activation? We don't have an answer on the molecular mechanism at this point, but Thr881 phosphorylation does not affect the level of pen-Thr phosphorylation (Fig. 3f, Supplementary Fig. 1c), nor does it affect its interaction with 14-3-3 proteins detected by Far Western analysis[27], suggesting that the effect of Thr881 phosphorylation on PM H⁺-ATPase

activation is not due to enhancing the interaction with the 14-3-3 protein, which is a known PM H⁺-ATPase activation mechanism. On the other hand, it will be necessary in future to confirm whether Thr881 phosphorylation affects the interaction between PM H⁺-ATPase and 14-3-3 protein in vivo. Other possibility is that the Thr881 phosphorylation might modify the conformation of the C-terminal autoinhibitory domain of PM H⁺-ATPase by promoting interaction with proteins other than 14-3-3 protein, thereby affecting activity promotion. By comparing the interacting proteins in AHA1-T881A and AHA1-T881D, it may be possible to find out the candidate proteins.

In summary, Fig. 6 represents a schematic model of light-induced stomatal opening and Thr881 phosphorylation of PM H⁺-ATPase in guard cells. The present results showed importance of Thr881 phosphorylation of PM H⁺-ATPase in guard cells for stomatal opening in *Arabidopsis thaliana*. Thr881 phosphorylation was regulated by both phototropin-mediated blue light-signaling and guard-cell photosynthesis, although pen-Thr phosphorylation of PM H⁺-ATPase was induced only by phototropin-mediated blue light signaling in isolated guard cells (Fig. 1). In previous works, we showed that red light irradiation of the intact leaves of *Arabidopsis thaliana* induced the phosphorylation of pen-Thr in guard cells and stomatal opening via photosynthesis in mesophyll cells[11,12]. The phosphorylation status of Thr881 in guard cells under red light illumination in intact leaves remains to be determined. Thr881 phosphorylation of PM H⁺-ATPase contributes around 50% compared to light-induced stomatal opening in Col-0 under our experimental conditions (Fig. 3c). In contrast, pen-Thr phosphorylation of PM H⁺-ATPase was essential for light-induced stomatal opening (Fig. 3i). Curiously, a mutant of BHP, a blue light-signaling component that interacts with BLUS1, showed constitutively high phosphorylation level of Thr881 in guard cells (Fig. 2). Phosphorylated Thr881 was dephosphorylated by PP2C.Ds as well as phosphorylated pen-Thr (Fig. 5). Identification of protein kinase for Thr881 in guard cells would provide the detailed molecular mechanisms of PM H⁺-ATPase activation and cooperation between phototropin-mediated blue light-signaling and guard-cell photosynthesis.

## Methods

### Plant materials and isolation of GCPs

As described in the previous reports[36,42], *Vicia faba* (broad bean; Ryosai Issun) was cultivated hydroponically in a greenhouse at 20 °C under sunlight. The culture solution containing 1/1000 Hyponex and 1/1000 Hoagland No.2 micronutrients were supplied with continuous aeration. GCPs from *Vicia faba* were isolated enzymatically from the lower epidermis of 4–6-week-old leaves as described elsewhere[6]. *Arabidopsis thaliana* plants were grown in soil under 16 h fluorescent light (50 μmol m⁻² s⁻¹)/8 h dark cycle at 24 °C in 55–70% humidity in a growth room. *Arabidopsis* Columbia-0 (Col-0) plants were used as the wild-type. Col-0 is a background ecotype of T-DNA insertion mutants of *aha1-9* (SAIL_1285_D12; At2G18960), *psy1r-1* (SALK_072802; At1g72300), *pskr1-2* (SAIL_673_H07; At2g02220), *pskr2* (SALK_024464; At5g53890), *pp2c.d2* (SALK_203806; At3g17090) and *pp2c.d6* (SAIL_171H03; At3g51370). The T-DNA insertion mutants were obtained from the *Arabidopsis* Biological Resource Center (Ohio State University). *gl1*, Columbia ecotype, is a background plant of *phot1-5 phot2-1*[7]. GCPs from *Arabidopsis thaliana* were isolated enzymatically from rosette leaves according to a previous method with modifications[36,43]. The epidermal tissues isolated from fully expanded 4–6-week-old *Arabidopsis* plants were treated with the first-step digestion medium (0.5% [w/v] Cellulase R-10 [FUJIFILM Wako Pure Chemical Corporation], 0.05% [w/v] Macerozyme R-10 [FUJIFILM Wako Pure Chemical Corporation], 0.1% (w/v) polyvinylpyrrolidone K-30, 0.2% (w/v) bovine serum albumin (BSA), 0.25 M mannitol, 1 mM CaCl₂, 10 mM MES-KOH, pH 5.4). For second digestion, epidermal tissues were treated with the digestion medium containing 1.5% (w/v)

Cellulase-RS (FUJIFILM Wako Pure Chemical Corporation), 0.02% (w/v) Pectolyase Y-23 (FUJIFILM Wako Pure Chemical Corporation), 0.2% (w/v) BSA, 0.4 M mannitol and 1 mM $CaCl_2$ (adjusted to pH 5.4). Collected GCPs were further purified with Histopaque-1077 (Sigma-Aldrich). Finally, purified GCPs were suspended in 0.4 M mannitol containing 1 mM $CaCl_2$ and kept on ice in the dark. Protein concentrations were determined using a Bradford kit according to the manufacturer's instructions (Bio-Rad Laboratories, Hercules, CA).

## Antibodies

Polyclonal rabbit anti-$H^+$-ATPase antibody and anti-pen-pThr antibody, raised against the catalytic domain of AHA2 and the synthetic phosphopeptide IETPSHYpTV from C-terminal 9 amino acid of AHA2, were used to detect PM $H^+$-ATPases and phosphorylated pen-Thr of PM $H^+$-ATPases, respectively[34]. Polyclonal anti-BHP antibody was used to detect BHP[10]. The polyclonal anti-pThr881 antibody against the synthetic phosphopeptide AQAQRpTLHGLQP from AHA1 was raised in rabbit.

Anti-$H^+$-ATPase antibody raised against the catalytic domain of AHA2 similarly recognized both AHA1 and AHA2, which are the main isoforms of PM $H^+$-ATPases expressed in *Arabidopsis* (Supplementary Fig. 1a). Accordingly, the signal intensity obtained with this antibody represented total PM $H^+$-ATPases. An immunoblot analysis of *Arabidopsis* GCPs using anti-pThr881 antibody (Supplementary Fig. 1b) showed that this antibody cross-reacted only with PM $H^+$-ATPases; thus, the immunohistochemical signals detected by the antibody corresponded only to PM $H^+$-ATPase. The anti-pen-pThr antibody cross-reacted with GFP-AHA1-WT, -T881A and -T881D but not with GFP-AHA1-T948A transiently expressed in MCPs, and the anti-pThr881 antibody cross-reacted with GFP-AHA1-WT and -T948A but not with GFP-AHA1-T881A and -T881D (Supplementary Fig. 1c). These results confirmed the specific detection by anti-pen-pThr and anti-pThr881 antibodies of the phosphorylated forms of pen-Thr and Thr881 in PM $H^+$-ATPases, respectively.

Mouse monoclonal anti-GFP (mouse $IgG_{1K}$, clones 7.1 and 13.1) and anti-FLAG antibodies (mouse $IgG_1$, clone M2) were obtained from Roche and Sigma-Aldrich.

## Phosphoproteomics

In the analysis of light responses, GCPs from *Vicia faba* (500 μg protein) were prepared in a suspension buffer (5 mM MES-NaOH [pH 6.0], 10 mM KCl, 0.4 M mannitol, and 1 mM $CaCl_2$). They were kept in the dark for 30 min or exposed to light. Strong red light (600 μmol $m^{-2}\,s^{-1}$) was provided for 30 min, followed by the superimposition of blue light (100 μmol $m^{-2}\,s^{-1}$) for 1 min. The response to FC was analyzed by treating GCPs from *Vicia faba* (200–700 μg protein) prepared in a suspension buffer with 10 μM FC for 5 min in the dark. The GCPs were disrupted by the addition of trichloroacetic acid to a final concentration of 20% (v/v), followed by centrifugation. The precipitated guard-cell proteins were suspended in a digestion buffer (8 M urea, 250 mM ammonium bicarbonate, 1× PhosSTOP [Roche]). The suspensions were reduced with Tris(2-carboxyethyl) phosphine hydrochloride, alkylated by iodoacetamide, and digested with Lys-C (FUJIFILM), followed by tryptic digestion. Digestions were performed with the enhancer ProteaseMAX™ Surfactant (Promega). The digested samples were acidified and desalted on MonoSpin C18 columns (GL Sciences). Phosphopeptides were enriched by immobilized metal affinity chromatography (IMAC, Agilent) from 100 μg of digested peptides and diluted with 0.1% (v/v) TFA, 2% (v/v) AcCN in distilled water for nano-LC-MS/MS. Nano-LC-MS/MS was performed using a Dionex U3000 Gradient Pump (Thermo Fisher Scientific) connected to a Q-Exactive Hybrid Quadrupole-Orbitrap Mass Spectrometer (Thermo Fisher Scientific). Enriched phosphopeptides were loaded onto a trap column (L-column ODS, 300 μm internal diameter [ID] × 5 mm, 5 μm particle size, CERI), and eluates were separated on a nano-HPLC capillary column (NTCC-360, 100 μm ID × 125 mm, 3 μm particle size, Nikkyo Technos).

Peptides and proteins were identified through automated database searching using Proteome Discoverer 2.2.0.388 (Thermo Fisher Scientific) against an expression database of *Vicia faba*. The following search parameters were applied: peptide mass range (m/z), 350–1800 Da; enzyme specificity, trypsin or Lys-C with up to two missed cleavages; precursor ion and peptide fragment mass tolerances, ±10 ppm and ±0.02 Da, respectively; static modification, carbamidomethyl (Cys); dynamic modifications, phosphorylation (Ser, Thr, and Tyr), and oxidation (Met). Peptide validation was conducted using the Percolator algorithm, and only high-confidence peptides were utilized for peptide identification and quantification. The resulting dataset, containing information on annotated sequences, modifications, master protein accession, peptide spectrum matches, and the total number of identified peptide spectra for each identified peptide, was imported into Microsoft Excel. Through the filter function in Excel, peptides with no phosphorylated residues were excluded from the list. The resulting phosphopeptides were then compared between the datasets.

## Measurement of the stomatal aperture

The stomatal aperture in epidermal tissues was measured as described previously[10], with minor modifications. Fully expanded rosette leaves were harvested from dark-adapted 4–5-week-old plants, with at least five plants in each experiment. The leaves were blended in a Waring blender (Waring Commercial) in 35 mL MilliQ water. The epidermal fragments were collected on a nylon mesh, rinsed with MilliQ water, and incubated in basal buffer (5 mM MES-bistrispropane, 10 mM KCl, and 0.1 mM $CaCl_2$, pH 6.5). Then they were irradiated with red/blue mixed light (i.e., blue light [Stick- B-32; EYELA, Tokyo, Japan] at 10 μmol $m^{-2}\,s^{-1}$ superimposed on background red light [LED-R; EYELA] at 50 μmol $m^{-2}\,s^{-1}$) at 24 °C or kept in the dark for 4 h. The stomatal aperture was measured microscopically in the abaxial epidermis by focusing on the inner lips of the stomata. Thirty stomatal apertures (five stomata per epidermal fragment) were measured accordingly in each independent experiment. All measurements of the stomatal response to light were made between 12:00 p.m. and 3:00 p.m.

## Gas exchange

The stomatal conductance of intact leaves was measured using an LI-6800 portable photosynthesis system (LI-COR) according to a modification of previous methods[3,29]. A leaf chamber (6800–12 A; LI-COR) was used in which the flow rate, leaf temperature, humidity, and ambient $CO_2$ concentration were kept at 500 μmol $s^{-1}$, 22.05 °C, 55%, and 400 μmol $mol^{-1}$, respectively. Red and blue light were provided by a light source (6800–02; LI-COR) attached to the chamber. The light response was measured after an initial 30 min dark adaptation. During light illumination, the total light intensity was kept at 600 μmol $m^{-2}\,s^{-1}$. Initially, the leaves were illuminated for 1 h only with red light (600 μmol $m^{-2}\,s^{-1}$). After saturated stomatal conductance by red light illumination, both blue light (60 μmol $m^{-2}\,s^{-1}$) and red light (540 μmol $m^{-2}\,s^{-1}$) were supplied. Blue light illumination was conducted for 30 min, followed by a return to red light illumination only.

## Western blotting analysis and immunohistochemistry

As described in the previous our report[6,34], immunoblot analyses were carried out using specific antibodies; anti-$H^+$-ATPase, anti-pen-pThr, anti-pThr881 and anti-BHP antibodies as a primary antibody, a goat anti-rabbit IgG conjugated to horseradishperoxidase (Bio-Rad Laboratories) as a secondary antibody. For detection of GFP and FLAG, a goat anti-mouse IgG conjugated to horseradishperoxidase (Bio-Rad Laboratories) was used as a secondary antibody. Immunohistochemical staining was performed as described previously with minor modifications[35]. In brief, epidermal fragments isolated from 4–5-week-old plants were illuminated with red light (50 μmol $m^{-2}\,s^{-1}$) for 20 min, after which a blue light pulse (10 μmol $m^{-2}\,s^{-1}$) was applied for 2.5 min.

Epidermal fragments were fixed just before or 2.5 min after blue light illumination.

## Vector construction for transgenic plants

A genomic DNA fragment of *AHA1* containing the promoter and terminator regions (4188 bp upstream from the start codon to 1106 bp downstream from the stop codon) was PCR-amplified using the specific primers gAHA1-Fw1 and gAHA1-Rv1 (Supplementary Table 4) and cloned into the *Hin*d III–*Xba* I site of *pCAMBIA1300* using the In-Fusion HD cloning kit (Clontech). The resultant plasmid was designated *pCAMBIA1300/gAHA1-WT*. *pCAMBIA1300/gAHA1-T881A*, *pCAMBIA1300/gAHA1-T881D*, and *pCAMBIA1300/gAHA1-T948A* were constructed by PCR-based site-directed mutagenesis using specific primers (Supplementary Table 4) and *pCAMBIA1300/gAHA1-WT* as the PCR templates. For example, for *pCAMBIA1300/gAHA1-T881A*, the two PCR products amplified using the respective primer sets (gAHA1-Fw2 and AHA1-T881A-Rv; AHA1-T881-Fw and gAHA1-Rv2) were cloned into the *Kpn* I site of *pCAMBIA1300/gAHA1-WT* using the In-Fusion HD cloning kit.

All constructed vectors were transformed into the *aha1−9* mutant by floral dipping using *Agrobacterium tumefaciens* GV3101.

## Vector construction for transient expression in MCPs

The transient expression of GFP-fused AHA1 in MCPs was achieved by constructing the vector *pUC18/p35S:mGFP::Linker::AHA1:nos3'*, in which monomerizing mutated GFP was fused to the N-terminus of AHA1 via a linker sequence [Gly$_4$Ser$^2$]. In *pUC18/p35S:mGFP::Linker::AHA1:nos3'*, three mutations (S65T, Q69M and A206K) relative to avGFP were made in *mGFP*, and an *Xho* I site was newly introduced into *AHA1* without altering its encoded amino acids (Thr634 and Ser635).

First, *pUC18/p35S:mGFP::Linker:nos3'* was constructed by seamlessly fusing the three DNA fragments PCR-amplified using three primer sets (Linker-Fw and mGFP-Rv1; mGFP-Fw1 and mGFP-Rv2; mGFP-Fw2 and Linker-Rv, see Supplementary Table 5) and *pUC18/p35S:sGFP(S65T):nos3'* as the template. Next, *pUC18/p35S:mGFP::Linker::AHA1-WT:nos3'* was constructed by seamlessly fusing two DNA fragments PCR-amplified using two primer sets (Linker-AHA1-Fw and AHA1-XhoI-Rv1; AHA1-XhoI-Fw1 and AHA1-XhoI-Rv2) and *pUC18/p35S:mGFP::Linker:nos3'* as the template into the *Xho*I site of *pUC18/p35S:mGFP::Linker:nos3'*. To construct *pUC18/p35S:mGFP::Linker::AHA1-T881A:nos3'*, *pUC18/p35S:mGFP::Linker::AHA1-T881D:nos3'*, and *pUC18/p35S:mGFP::Linker::AHA1-T948A:nos3'*, PCR products amplified with *pUC18/p35S:mGFP::Linker::AHA1-WT:nos3'* as the template and two primer sets (AHA1-XhoI-Fw2 and AHA1-T881A-Rv; AHA1-T881-Fw and AHA1-WT-Rv), two primer sets (AHA1-XhoI-Fw2 and AHA1-T881D-Rv; AHA1-T881-Fw and AHA1-WT-Rv), and one primer set (AHA1-XhoI-Fw2 and AHA1-T948A-Rv2), respectively, were seamlessly fused to the *Xho* I site of *pUC18/p35S:mGFP::Linker::AHA1-WT:nos3'*.

*pUC18/p35S:sGFP(S65T)::PP2C-C4* and *pUC18/p35S:sGFP(S65T)::PP2C-D6* were constructed in a previous study[29].

## Transient expression analysis using Arabidopsis MCPs

Transient expression in MCPs was analyzed according to previous methods[29,44]. MCPs prepared from the rosette leaves of 3–5-week-old *Arabidopsis* seedlings (Col-0) were co-incubated with the plasmid in PEG solution (final concentration: 15% [w/v]) for 15 min, washed with incubation solution (1 mM Mes-KOH, pH 6.0, 0.4 M mannitol, 20 mM KCl, and 1 mM CaCl$_2$), and then incubated in the dark overnight. Then they were concentrated by centrifugation at $100\,g$ for 1 min for subsequent use in immunoblot analyses.

## Auxin- and FC-induced PM H$^+$-ATPase phosphorylation in hypocotyls

As described in the previous our report, auxin- and FC-induced PM H$^+$-ATPase phosphorylation were analyzed as described in a previous report[17]. Briefly, hypocotyl segments from 3-day-old etiolated seedlings were pre-incubated on growth medium (1 mM Mes-KOH, pH 6.0, 10 mM KCl, 0.8% agar) for 30−90 min and then transferred for 30 min onto growth medium containing the auxin 10 μM indole-3-acetic acid or 10 μM fusicoccin. Proteins from the collected hypocotyl segments were extracted with SDS-PAGE sample buffer and subjected to SDS-PAGE and to immunoblot analysis using anti-H$^+$-ATPase, anti-pen-pThr, and anti-pThr881 antibodies.

## Accession numbers

Sequence data from this article can be found in the Arabidopsis Information Resource website (https://www.arabidopsis.org/) under the following accession numbers: AHA1 (AT2G18960), AHA2 (AT4G30190), BLUS1 (AT4G14480), BHP (AT4G18950), PSY1R (AT1G72300), PSKR1 (AT2G02220), PSKR2 (AT5G53890), SAUR19 (AT5G18010), TUB2 (AT5G62690), PP2C.D1 (AT5G02760), PP2C.D2 (AT3G17090), PP2C.D3 (AT3G12620), PP2C.D4 (AT3G55050), PP2C.D5 (AT4G38520), PP2C.D6 (AT3G51370), PP2C.D7 (AT5G66080), PP2C.D8 (AT4G33920), PP2C.D9 (AT5G06750), PP2C.C4 (AT1G07630). The nomenclature of PP2C.Ds is according to Spartz et al.[38].

## Reporting summary

Further information on research design is available in the Nature Portfolio Reporting Summary linked to this article.

## Data availability

The raw MS data have been deposited in the ProteomeXchange Consortium via the PRIDE partner repository under accession numbers, PXD040016 for phosphoproteomics of light-illuminated GCPs and PXD040015 for phosphoproeomics of FC-treated GCPs. Source data are provided with this paper.

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

## Acknowledgements

We thank Professor Yoshikatsu Matsubayashi (Nagoya University) for providing *psy1r pskr1 pskr2* mutant seeds and Dr. Eigo Ando (University of Tokyo) for discussing immunohistochemical detection in guard cells. We also thank Ms Kyomi Taki (Nagoya University) for technical assistance and Dr. Emi Mishiro for technical advice on proteomic analyses. This work was supported by grants from the Ministry of Education, Culture, Sports, Science, and Technology, Japan (grant nos. 20H05687 and 20H05910 to T.K., 20K06685 to K.T.) and grant-in-aid for Japan Society for the Promotion of Science Research Fellow (grant no. 202214777 to K.F.).

## Author contributions

Y.H. and T.K. conceived the research project and designed the experiments. Y.H., K.F., K.T., S.N.K., K.Ka., T.S., K.Ku. and T.K. conducted the experiments and analyzed the data. All authors participated in the entire study and in writing the manuscript.

## Competing interests

The authors declare no competing interests.
