## [Peer Review File—NEW · Nature Communications]

Phosphorylation of plasma membrane H⁺ -ATPase Thr881 participates in light-induced stomatal openingReviewer #1 (Remarks to the Author):

The authors reported a systemic study that investigates the functional roles of two phosphorylation sites of AHA1 in controlling light-induced Arabidopsis stomatal opening. The topic is interesting and not well studied, but could be very relevant in our constantly changing environment. The genetic and physiological analyses all appear sound. The manuscript is well written, although there are some structural decisions and presentations which make the narrative slightly disjointed.

I am quite curious that the authors performed phosphoproteomics screen in *Vicia faba*, while all the functional validation of AHA1 was studied in Arabidopsis. Although the authors explained the experimental design in the main text from line 269 to line 277, such interpretation seems a bit farfetched as they are totally different plant species and this is not real *in vivo* analysis. In this case, I think the fundamental rationale is not correct. In addition, how the data were processed and how phosphopeptides were normalized against non-phosphorylated peptides (no information presented)? The reviewer would suggest to directly investigate the phosphorylation changes of AHA1 using Arabidopsis leaf (membrane) phosphoproteomics. Definitely the phosphoproteomics analysis in *Vicia faba* stomatal can give some hints about the molecular behaviors of AHA or other key proteins that are directly involved in stomatal opening, but the final study should go to Arabidopsis since current mass spectrometry allows to measure over thousands of phosphoproteins (including AHAs).

In conclusion, as I said I think the work is generally fine but the authors should consider a more solid/accurate analysis in Arabidopsis phosphoproteomics and make it more relevant to the topic of this research.

Reviewer #2 (Remarks to the Author):

Resume: The first part of the manuscript reports the phosphorylation residues detected (by phospho-proteomic approaches) on H⁺-ATPases from *Vicia faba* guard cell protoplasts, after appropriate stimulations. Besides the penultimate T948 (which is well-known to be phosphorylated when the pump is active), several other amino acids, particularly T881 were phosphorylated as well in responding to blue and red light illumination. Both T881 and T948 were also phosphorylated after treatment with the fungal toxin, fusicoccin or auxin (on hypocotyl segments).

The second part of the manuscript presented functional characterization with Arabidopsis. GFP-tagged AHA1, either wild type, or specifically mutated (e.g. T881A, T881D, and T948A) were examined by transient expression in protoplasts prepared from mesophyll cells. The authors possess antibodies able to distinguish H⁺-ATPases phosphorylated at T881 or T948. And so the authors recount: Auxin was reported to be a signal that can induce phosphorylation at T948 and T881. In contrast, fusicoccin reduced phosphorylation at T881. Red light only induced phosphorylation at T881, but not T948. The responsible T881 kinase, previously proposed by another lab, to be PSY1R, was not confirmed in this study. The authors showed, by the use of DCMU (binds to plastoquinone, the ultimate electron acceptor in PSII), that phosphorylation at T881 requires photosynthesis. T881 phosphorylation seems constitutive in the *bhp* (a RAF-like kinase) mutant background. In sum, T881 phosphorylation may be triggered by a number of environmental signals (lines 203 to 205). In GFP-AHA1-T881A and T-881D, pen-T948 was still phosphorylated, while in GFP-AHA1-T948A, T881 was also phosphorylated. Therefore, phosphorylation at the two sites seems, in some cases, independent. Previous work has identified PP2C.D isoforms in the dephosphorylation of pen-T948. Both of these phosphorylated residues were targets of dephosphorylation by PP2C.D isoforms. In mutants impaired in several PP2C isoforms, the phosphorylation levels were higher at T881 and T948.

The manuscript is very descriptive. It is fine to get a panoramic view of the type of stimuli that are correlated with phosphorylation of T881 and T948, if a reader is not asking for mechanistic understanding.

Some comments and questions:

1. Add the year to the reference Amano et al. (reference 38).

2. Line 46. "...causes guard cell apertures to swell and thus enlarge." May be clearer to non-specialists if written along the line: the flanking guard cells swell, and their arcing away from each other enlarges the stomatal aperture. Or something to this line.

3. This referee is not sure of the novelty of the results concerning T881 being detected as a phosphorylatable site because the authors have cited previous work in yeast. So this paper can be considered a refinement guard cells. But anything that can activate the proton pumping activity will likely lead to stomatal opening, a proof of principle experiment. Moreover, it looks like auxin can also stimulate T881 phosphorylation, as low K⁺, so light adds to the list of stimuli.

4. Have authors any idea whether some of the SnRKs that have been implicated in ion channel regulation might be involved in the phosphorylation of T881?

5. Line 93. Claim that this is the first case that T881 phosphorylation in plant physiological responses. But there was also the mention in the submission that T881 phosphorylation is also stimulated by low K⁺, attributed to Wang et al. 2022 (reference 30). There is also Li et al. (2021) on auxin response, in which T881 in AHA1 or equivalent site in AHA2 were also identified as phosphorylated, although auxin does not evoke stomatal opening.

Need some re-phrasing. But I am not sure whether being first brings any value to the work.

6. Any idea(s) for the photosynthate(s) that stimulate(s) phosphorylation at T881? The notion that the mesophyll cells repond to light and send a signal to guard cells has already been proposed since several decades. And the use of DCMU to show the link between photosynthesis and the red light-induced stomatal response has also been reported independently by several labs, including some of the same authors, about 20 years ago.

7. Any idea why the phosphorylation level becomes constitutive at T881 in the bhp mutant background?

8. Why neither the T881D nor pen-T948D mutation in the AHA1 transgene restores stomatal opening response in the aha1-9 background is curious. In fact, none of the engineered mutants tested restored H⁺-ATPase activity in the aha1-9 background. If the reader has not been pre-conditioned by the authors' focus on phosphorylation, these negative results actually invite many other possible interpretations, including protein local conformation by H-bonding.

Response to reviewers

We appreciate the constructive suggestions of the reviewers. We revised the manuscript in accordance with the suggestions to include additional experiments and analyses. In the revised manuscript, we added Mr. Taku Sakakibara as our co-author due to his contribution during the revision process. In addition, we found a mistake regarding *pp2c.d* mutant in Figure 4C. We only used *pp2c.d2d6* double mutant. So, we modified Figure 4C. Below, we respond to the reviewers' comments point by point.

Black: Referee comments.

Blue: Authors' response.

Cyan highlight: Authors' modification in the manuscript.

REVIEWER COMMENTS

Reviewer #1 (Remarks to the Author):

The authors reported a systemic study that investigates the functional roles of two phosphorylation sites of AHA1 in controlling light-induced Arabidopsis stomatal opening. The topic is interesting and not well studied, but could be very relevant in our constantly changing environment. The genetic and physiological analyses all appear sound. The manuscript is well written, although there are some structural decisions and presentations which make the narrative slightly disjointed.

I am quite curious that the authors performed phosphoproteomics screen in *Vicia faba*, while all the functional validation of AHA1 was studied in Arabidopsis. Although the authors explained the experimental design in the main text from line 269 to line 277, such interpretation seems a bit farfetched as they are totally different plant species and this is not real in vivo analysis. In this case, I think the fundamental rationale is not correct.

Response:

We would like to thank Reviewer #1 for taking the time to read and comment on our manuscript and providing a thoughtful review.

Phosphorylation of T881 of plasma membrane (PM) H⁺-ATPase in guard cells was initially found in guard cell protoplasts (GCPs) from *Vicia faba*, because GCPs can be obtained at higher purity and in larger amounts than from *Arabidopsis thaliana*. However, we confirmed that a similar response was observed in guard cells from *Arabidopsis thaliana* by immunohistochemical analysis (Fig. 1) and Western blotting (Supplementary Fig. 2). As you pointed out, the experiments on phosphoproteomics were performed on a different plant, *Vicia faba*. Thus, we conducted a new phosphoproteomics study using Arabidopsis GCPs. The results confirmed that, similar to *Vicia* GCP, the phosphorylation levels of the Thr881 of major PM H⁺-ATPase isoforms, AHA1 and AHA2, significantly increased in response to blue light on three different occasions (Table 1 in Response letter). However, due to the limitation of sample amount, we examined effects of red and blue light, but not the effect of darkness on phosphorylation levels. Furthermore, the present data contain many new and interesting data other than PM H⁺-ATPase. We'd like to investigate these new components. Therefore, we would prefer not to provide phosphoproteomics data from Arabidopsis GCPs in the present manuscript.

Table 1 in Response letter

Phospho-site	Abundance (x 10 ⁵)									
	Rep. 1			Rep. 2			Rep. 3			
	R	R+B	Fold Change ([R+B]/[R])	R	R+B	Fold Change ([R+B]/[R])	R	R+B	Fold Change ([R+B]/[R])	
Thr881	AHA1	360	826	2.30	779	1642	2.11	331	883	2.67
	AHA2	18	28	1.53	66	145	2.20	8	22	2.69
	AHA1/ AHA2	29	51	1.74	571	932	1.63	352	722	2.05
Thr948	AHA1	2191	4286	1.96	2961	7168	2.42	2977	8892	2.99
	AHA2	569	976	1.72	563	1404	2.50	573	1646	2.87

In addition, how the data were processed and how phosphopeptides were normalized against non-phosphorylated peptides (no information presented)?

Response:

Thank you for your important suggestion. We apologize for the lack of information about normalization. In fact, for phosphoproteomics using GCP, the amount of protein that can be obtained is limited, so phosphopeptides were not normalized against non-phosphorylated peptides, although all samples are subjected to MS analysis starting with an equal amount of protein. Instead, we confirmed that there was no change in the amount of H⁺-ATPase protein in the dark, red light, or blue light in Western blotting and immunostaining using anti-H⁺-ATPase antibodies. We added these results to Fig1a and Supplementary Fig 2 and described on p.7, line 134, as follows:

in guard cells from *Arabidopsis* epidermis **without changing PM H⁺-ATPase amount** (Fig. 1a),

The reviewer would suggest to directly investigate the phosphorylation changes of AHA1 using *Arabidopsis* leaf (membrane) phosphoproteomics. Definitely the phosphoproteomics analysis in *Vicia faba* stomatal can give some hints about the molecular behaviors of AHA or other key proteins that are directly involved in stomatal opening, but the final study should go to *Arabidopsis* since current mass spectrometry allows to measure over thousands of phosphoproteins (including AHAs).

Response:

As you pointed out, it is very important to directly investigate the phosphorylation of AHA1 by phosphoproteomics using *Arabidopsis* GCPs. Thus, as described above, we conducted a new phosphoproteomics study using *Arabidopsis* GCPs. The results confirmed that, similar to *Vicia* GCP, the phosphorylation levels of the Thr881 of major PM H⁺-ATPase isoforms, AHA1 and AHA2, significantly increased in response to blue light on three different occasions (Table 1 in Response letter). However, due to the limitation of sample

amount, we examined effects of red and blue light, but not the effect of darkness on phosphorylation levels. Furthermore, the present data contain many new and interesting data other than PM H⁺-ATPase. Therefore, we would prefer not to provide phosphoproteomics data from Arabidopsis GCPs in the present manuscript.

In conclusion, as I said I think the work is generally fine but the authors should consider a more solid/accurate analysis in Arabidopsis phosphoproteomics and make it more relevant to the topic of this research.

Response:

In accordance with important your suggestions, we added the data regarding the amount of PM H⁺-ATPase protein. Furthermore, we performed a comprehensive phosphoproteomics analysis using Arabidopsis GCPs and confirmed phosphorylation changes at Thr881 of AHA1. We believe the revised manuscript has been improved greatly by your important comments.

Reviewer #2 (Remarks to the Author):

Resume: The first part of the manuscript reports the phosphorylation residues detected (by phospho-proteomic approaches) on H⁺-ATPases from *Vicia faba* guard cell protoplasts, after appropriate stimulations. Besides the penultimate T948 (which is well-known to be phosphorylated when the pump is active), several other amino acids, particularly T881 were phosphorylated as well in responding to blue and red light illumination. Both T881 and T948 were also phosphorylated after treatment with the fungal toxin, fusicoccin or auxin (on hypocotyl segments).

The second part of the manuscript presented functional characterization with *Arabidopsis*. GFP-tagged AHA1, either wild type, or specifically mutated (e.g. T881A, T881D, and T948A) were examined by transient expression in protoplasts prepared from mesophyll cells. The authors possess antibodies able to distinguish H⁺-ATPases phosphorylated at T881 or T948.

And so the authors recount:

Auxin was reported to be a signal that can induce phosphorylation at T948 and T881. In contrast, fusicoccin reduced phosphorylation at T881. Red light only induced phosphorylation at T881, but not T948. The responsible T881 kinase, previously proposed by another lab, to be PSY1R, was not confirmed in this study. The authors showed, by the use of DCMU (binds to plastoquinone, the ultimate electron acceptor in PSII), that phosphorylation at T881 requires photosynthesis. T881 phosphorylation seems constitutive in the bhp (a RAF-like kinase) mutant background. In sum, T881 phosphorylation may be triggered by a number of environmental signals (lines 203 to 205).

In GFP-AHA1-T881A and T-881D, pen-T948 was still phosphorylated, while in GFP-AHA1-T948A, T881 was also phosphorylated. Therefore, phosphorylation at the two sites seems, in some cases, independent.

Previous work has identified PP2C.D isoforms in the dephosphorylation of pen-T948. Both of these phosphorylated residues were targets of dephosphorylation by PP2C.D isoforms. In mutants impaired in several PP2C isoforms, the phosphorylation levels were higher at T881 and T948.

The manuscript is very descriptive. It is fine to get a panoramic view of the type of stimuli that are correlated with phosphorylation of T881 and T948, if a reader is not asking for mechanistic understanding.

Response:

We thank Reviewer #2 for taking the time to read and review our paper. We appreciate the thoughtful comments and careful review.

Some comments and questions:

1. Add the year to the reference Amano et al. (reference 38).

Response:

We are very sorry for our mistakes. We added the year to the reference in the revised manuscript.

2. Line 46. "...causes guard cell apertures to swell and thus enlarge." May be clearer to non-specialists if written along the line: the flanking guard cells swell, and their arcing away from each other enlarges the stomatal aperture. Or something to this line.

Response:

Thank you for your suggestion. We changed the description according to your suggestion, on Page 3, line 46-47, as follows.

The accompanying uptake of the water causes the flanking guard cells swell, and their arcing away from each other enlarges the stomatal aperture.

3. This referee is not sure of the novelty of the results concerning T881 being detected as a phosphorylatable site because the authors have cited previous work in yeast. So this paper can be considered a refinement guard cells. But anything that can activate the proton pumping activity will likely lead to stomatal opening, a proof of principle experiment. Moreover, it looks like auxin can also stimulate T881 phosphorylation, as low K⁺, so light adds to the list of stimuli.

Response:

Thank you for your comment. As the Reviewer#2 described, it has been shown in yeast systems that Thr881 is a phosphorylation site, and that Thr881 phosphorylation enhances plant PM H⁺-ATPase activity. However, the characterization of Thr881 in PM H⁺-ATPases was only examined in the heterogeneous organism of yeast, not in plant cells. That is to say, these insights in yeast systems suggest that T881 phosphorylation promotes PM H⁺-ATPase activity also in plant cells, but it is still unclear whether some environmental stimuli enhance Thr881 phosphorylation in plant cells and whether Thr881 phosphorylation leads to physiological responses. Although Wang et al. (2022) investigated relationship between low K⁺ and the phosphorylation level of Thr881, but there is no significant difference (Wang et al. 2022; Fig. 6H) and that the genetic analysis such as T881A mutant is lack to clarify the physiological role of Thr881 phosphorylation in their paper.

In our study, we demonstrated that blue and red light-induced Thr881 phosphorylation in guard cells by phosphoproteomics, and western blot and immunohistochemical analysis using the specific antibody against phospho-Thr881. We also genetically demonstrated that Thr881 phosphorylation exerts positive function in stomatal opening using AHA1-T881A mutants. Thus, we believe that this paper clearly demonstrated the significance of Thr881 phosphorylation for physiological responses in plant cells, especially for stomatal opening in guard cells.

4. Have authors any idea whether some of the SnRKs that have been implicated in ion channel regulation might be involved in the phosphorylation of T881?

Response:

Thanks for your interesting suggestion, to investigate whether SnRK2 is involved in the phosphorylation of Thr881, we examined the phosphorylation responses of PM H⁺-ATPase in GCP from the *snrk2.6/srk2e/ost1* mutant, the major SnRK2 in guard cells. Phosphorylation level of Thr881 tended to be slightly lower in the *ost1* mutant, but the response was very clear on three different occasions (Figure 1 in Response letter), suggesting that SnRK2s have no effect on light-induced Thr881 phosphorylation. Therefore, we would like to show these data only in Response letter.

Figure 1 in Response letter: Phosphorylation of PM H⁺-ATPase in Col-0 and *ost1-3*. GCPs were isolated from Col-0 and *ost1-3* and kept in the dark for 1 hr. Dark-adapted GCPs were illuminated with red light (R: 300 $\mu\text{mol m}^{-2} \text{s}^{-1}$, 20 min), then blue light (RB: 100 $\mu\text{mol m}^{-2} \text{s}^{-1}$, 2.5 min) was superimposed on the red light. The phosphorylated PM H⁺-ATPase and amount of PM H⁺-ATPase were detected by the immunoblot using anti-pThr881, anti-pen-pThr and anti-H⁺-ATPase antibodies. Upper panel shows typical results. Arrowheads show the position of AHAs. Lower graph shows the relative phosphorylation level of PM H⁺-ATPase, which is calculated by dividing the signal intensities of the bands detected with anti-pThr881 or anti-pen-pThr by that of the band detected with anti-H⁺-ATPase antibody. Data are mean \pm SD (pen-pThr) or exact value (pThr881) of three independent experiments. The asterisk indicates statistically significant difference (one-tailed Student's *t* test: *, $P < 0.05$; n.s., Not significant, $P > 0.05$).

5. Line 93. Claim that this is the first case that T881 phosphorylation in plant physiological responses. But there was also the mention in the submission that T881 phosphorylation is also stimulated by low K⁺, attributed to Wang et al. 2022 (reference 30). There is also Li et al. (2021) on auxin response, in which T881 in AHA1 or equivalent site in AHA2 were also identified as phosphorylated, although auxin does not evoke stomatal opening. Need some re-phrasing. But I am not sure whether being first brings any value to the work.

Response:

Thank you for your important comment. Actually, Thr881 phosphorylation in planta is encapsulated in the comprehensive data presented by several phosphoproteomic analyses, including Li et al (2021). However, there is no description regarding Thr881 phosphorylation in their paper. As we described above, low K⁺ tended to increase the phosphorylation level

of Thr881, but there is no significant difference (Wang et al. 2022). Therefore, we believe that this paper describes, for the first time, elevation of Thr881 phosphorylation level in response to environmental stimuli. On the other hand, as Reviewer#2 points out, "the first detection" of Thr881 phosphorylation in planta is not the highlight of this study, nor should be so strongly argued, so we'd like to delete the sentence in Line 93-94.

6. Any idea(s) for the photosynthate(s) that stimulate(s) phosphorylation at T881? The notion that the mesophyll cells repond to light and send a signal to guard cells has already been proposed since several decades. And the use of DCMU to show the link between photosynthesis and the red light-induced stomatal response has also been reported independently by several labs, including some of the same authors, about 20 years ago.

Response:

We thank you for the important suggestion. We don't know whether the signal from mesophyll cells is involved in Thr881 phosphorylation. However, at least, Thr881 phosphorylation depends on photosynthesis in guard cells. It is possible that Thr881 phosphorylation is also induced by the signal from mesophyll cells in response to red light, as in the case of pen-Thr (Thr948). We would like to investigate this possibility in the future by using the whole leaf system. But, as shown in Ando and Kinoshita (2018) Plant Physiology, the phosphorylation of Thr948 of PM H⁺-ATPase in guard cells is not induced by exogenous sucrose. This may not be the case for sucrose as a photosynthate. We would like to further analyse this point.

7. Any idea why the phosphorylation level becomes constitutive at T881 in the bhp mutant background?

Response:

We are very interested in it, but we do not currently have any ideas. We would like to carry out further analyses, for example, to search for BHP interaction proteins in guard cells.

8. Why neither the T881D nor pen-T948D mutation in the AHA1 transgene restores stomatal opening response in the aha1-9 background is curious. In fact, none of the engineered mutants tested restored H⁺-ATPase activity in the aha1-9 background. If the reader has not been pre-conditioned by the authors' focus on phosphorylation, these negative results actually invite many other possible interpretations, including protein local conformation by H-bonding.

Response:

As shown in Supplementary Figure 5, in T881D plants, there is a tendency to promote light-induced stomatal opening, although it's not significantly, and it is possible that the activity is partially restored, and further detailed analysis is required. As you know, phosphorylation of pen-Thr (Thr948) is thought to create high-affinity binding site for the 14-3-3 protein, causing a displacement of the C-terminal region and activation of the PM H⁺-ATPase.

Although further analysis is required, it is speculated that T948D is different from the true phosphorylation and therefore cannot be the high-affinity binding site for the 14-3-3 protein.

Reviewer #1 (Remarks to the Author):

1. In the 2nd submission, the author insists that phosphoproteomics analysis of *Vicia faba* is consistent with that of *Arabidopsis*, in this regard the author did not provide phosphoproteomics data from *Arabidopsis* GCPs. OK, considering this work is focused on the functional characterization of AHA, it is fine. Although this work analyzed a lot about light induced phosphorylation changes of T948 and T881 using different mutants, all analysis were performed at physiological level and the conclusion remains descriptive. The author demonstrated that PP2C.D dephosphorylate Thr881, which did not provide novel findings in the present version as this has been reported in previously published literature. As showed in the last model, authors did not provide evidence for the upstream kinase (or regulator proteins) of these two phosphorylation sites, most experiments shows physiological changes. In this regard, the work did not explore deeply for the molecular functions of targeted gene

Reviewer #3 (Remarks to the Author):

Dear Authors and Editor,

Regulation of plasma membrane proton pumps in guard cells is one of the main factors by which plants modulate their stomatal conductance. In turn, the stomatal conductance is a main determinate for CO₂ uptake and water loss, and thus very important for plants to adapt to changes in environmental growth conditions.

In previous papers, Toshinori Kinoshita and his group showed that the H⁺-ATPases of guard cells are regulated through a phosphorylation of the penultimate Threonine in the C-terminus of the pump. In the submitted manuscript they now report on a second phosphorylation site (T881) in the C-terminus, which also appears to be involved in modulation of H⁺-ATPase activity. The data show that T881 gets phosphorylated in response to red- and blue light and enhances the light-induced stomatal opening. Moreover, the authors show that T881 is not only involved in regulation of H⁺-ATPases in guard cells, but also in mesophyll cells and hypocotyls in response to auxin. I regard it very likely that this study will attract the attention of many researchers in the field and stimulate further research.

The data presented in the manuscript are in general clear and consistent, with the exception of stomatal conductance, which should be presented as absolute values, as explained I the major points of concern. Unfortunately, the result- and discussion sections are not easy to understand and fail to explain the logic of the experimental approach. To my opinion these sections should be reworked in such a way that also non-expert scientists can read and understand the work.

Major points of concern:

1. The stomatal conductance data in Fig. 2d are given as "relative value to gsw", but it is not explained how these values were calculated and why the data are presented in this way. It is common to give stomatal conductance as absolute values of the water conductance, relative to the leaf surface. This would enable the reader to see, if the conductance of the plant lines already differs at the start of the experiment. Note that plants with wide open stomata are likely to show a smaller increase in stomatal conductance in response to light, as compared to the control. Moreover, the proper control in Fig. 2d would be the *aha1-9* mutant, complemented with the non-modified AHA1 gene. Please include the stomatal conductance response of this line in this graph.
2. In the discussion (line 275) it is mentioned that genomic data for *Vicia faba* are missing. This comes as a surprise to me, as the genome of broad bean has been sequenced and the data were published in Jayakodi et al., 2023 (Nature <https://doi.org/10.1038/s41586-023-05791-5>).
3. The bars in the graph of Figure S5d seem to be colored the wrong way around. The shorter bars should probably be red and the longer bars blue.

Below I will point out a number of points that need attention, but I think that in general the

authors need to rework most of the text in the results and discussion.

1. In the introduction, one can get the impression that the authors did not show that the phosphorylation of T881 is also enhanced by red- and blue light in Arabidopsis guard cells. Please modify the text, so the reader understands that light-induced phosphorylation of T881 in guard cells is conserved for broad bean and Arabidopsis.
2. Results lines 100-112. The first sentences of the results are a repetition of information that is already given in the introduction. It would make more sense that the authors explain that they performed a phosphoproteomics analysis with guard cells of *V. faba*, because of the high quality and quantity of phosphoproteins that they can obtain with this species. In the manuscript this information is given at the start of the discussion, but it would make more sense to explain the choice for *V. faba* at the start of the results.
3. Results lines 113-124. The impact of red- and blue light on phosphorylation on the positions T881 and T948 can be explained more clearly. As far as I understand, both red- and blue light promote phosphorylation of T881, whereas the phosphorylation at T942 is only induced by blue light in isolated epidermis and guard cells protoplasts. This could be explained more clearly.
4. Results lines 126-139. How did the authors conduct the experiments with Arabidopsis stomata. Did they use epidermal fragments, isolated epidermal peels, or intact leaves. This is not well explained and because of this, the reader does not get a good feeling of the conditions used for these experiments.
5. Results lines 148-152. Why are the results with the bhp-1 mutant shown in the supplementary figures? The impact of loss of BHP on the phosphorylation status of T881 is unexpected and show that BHP is not required to phosphorylate this position of the H⁺-ATPases.
6. Results lines 153-156. Here phosphoprotein data obtained with *V. faba* guard cells are mentioned, which are not related to the previous text. These data could be better be moved to the start of the results in which the *V. faba* phosphoproteome data are discussed.
7. Results line 166. Here it is written that transgenic plants carrying wild type AHA1 in *aha1-9* show the same light-induced stomatal opening as wild type (Figure S4). Does this mean that the experiment in Figure S4 was carried out with intact plants?
8. Results lines 193-217. In the first sentence of the section, it is explained that red light causes phosphorylation of the T948 in intact leaves, but the remaining of the text does not fit to this. The authors should explain why they tested the impact of light on the phosphorylation of AHA proteins in mesophyll, and the phytohormone auxin in hypocotyl cells.
9. Results lines 234-250. Please give a reference that explains the nomenclature of the PP2C proteins.
10. Discussion, lines 288-293. Please discuss the unexpected results with the bhp-1 mutant in further detail. Does the phosphorylation of T881 fit to the phenotype of this mutant?
11. Discussion, lines 336-338. Here it is written that "T881 phosphorylation of the PM-H⁺-ATPase contributes around 50% to ...". How did the authors calculate this number and to which conditions do they refer?
12. Figure 5. It should be mentioned in the legend that this scheme reflects the situation of isolated stomata, as or intact leaves red light also promotes phosphorylation of T948.

Reviewer #4 (Remarks to the Author):

The manuscript entitled "Phosphorylation of plasma membrane H⁺-ATPase Thr881 participates in light-induced stomatal opening", describes the AHA1 phosphorylation of Thr881 and its role in PM H⁺-ATPase function in stomatal guard cells in response to red and blue light. The authors conclude that Thr881 phosphorylation by both red and blue light influences plasma membrane AHA1 function in stomatal guard cells and contributes to stomatal opening. The results suggest that this phosphorylation event is regulated by phototropin-mediated blue light-signaling and guard-cell

photosynthesis. This manuscript has interesting results but falls short in demonstrating their results and several conclusions are too speculative. Also, I fully support the assessments made by the two referees, particularly with regard to two crucial points.

Firstly, the decision to perform a phosphoproteomic screen in *Vicia faba* while restricting the functional validation of AHA1 (vs AHA2) exclusively to *Arabidopsis* raises notable concerns. Although the authors' attempt to clarify the experimental design is commendable, the interpretation seems somewhat strained, since there are inherent differences between these plant species. For example, FC treatment did not induce the same effects on T881 phosphorylation in *Arabidopsis thaliana* and *Vicia faba* (Fig. 1 and Supplementary Table 3). This deviation from a proper *in vivo* analysis raises questions about the rationale for this approach.

Secondly, I fully agree with the observation that the manuscript tends to be highly descriptive. While obtaining a panoramic view of the stimuli correlated with phosphorylation of T881 and T948 is valuable, the absence of a more in-depth mechanistic investigation may reduce the overall impact of the manuscript. In particular, the authors discuss potential candidates for protein kinases involved in the phosphorylation of Thr881 and raise questions about the mechanism by which red light induces this phosphorylation. Striking a balance between presenting an overview and delving deeper into mechanisms could improve the comprehensiveness of the manuscript and make it more appealing to a wider audience. Consider incorporating more in-depth analyses to strengthen the scientific basis of the manuscript and increase its relevance.

Response to reviewers

We appreciate the constructive suggestions and comments of the reviewers. We revised the manuscript in accordance with the suggestions and comments.

Black: Reviewer comments.

Blue: Authors' response.

REVIEWERS' COMMENTS

Reviewer #1 (Remarks to the Author):

1. In the 2nd submission, the author insist that phosphoproteomics analysis of *Vicia faba* is consistent with that of *Arabidopsis*, in this regard the author did not provide phosphoproteomics data from *Arabidopsis* GCPs. OK, considering this work is focused on the functional characterization of AHA, it is fine.

Response:

Thank you for your understanding.

Although this work analyzed a lot about light induced phosphorylation changes of T948 and T881 using different mutants, all analysis were performed at physiological level and the conclusion remains descriptive.

Response:

We agree with your opinion. We'd like to clarify the molecular mechanism for light-induced phosphorylation of H⁺-ATPase in guard cells by future experiments.

The author demonstrated that PP2C.D dephosphorylate Thr881, which did not provide novel findings in the present version as this has been reported in previously published literature.

Response:

We think this is misunderstanding of Reviewer#1. Dephosphorylation of phosphorylated Thr881 by PP2C.D is a novel finding. Previously, we showed dephosphorylation of phosphorylated Thr948 by PP2C.D in guard cells (Akiyama et al. *Plant Physiol.* 2022).

As showed in the last model, authors did not provide evidence for the upstream kinase (or regulator proteins) of these two phosphorylation sites, most experiments shows physiological changes. In this regard, the work did not explore deeply for the molecular functions of targeted gene

Response:

We agree with your opinion. But, in this study, we found importance of phosphorylation of Thr881 in response to red and blue light in guard cells for light-induced stomatal opening and its function. Further experiments would provide the detailed molecular mechanism of this responses.

Reviewer #3 (Remarks to the Author):

Dear Authors and Editor,

Regulation of plasma membrane proton pumps in guard cells is one of the main factors by which plants modulate their stomatal conductance. In turn, the stomatal conductance is a main determinate for CO₂ uptake and water loss, and thus very important for plants to adapt to changes in environmental growth conditions.

In previous papers, Toshinori Kinoshita and his group showed that the H⁺-ATPases of guard cells are regulated though a phosphorylation of the penultimate Threonine in the C-terminus

of the pump. In the submitted manuscript they now report on a second phosphorylation site (T881) in the C-terminus, which also appears to be involved in modulation of H⁺-ATPase activity. The data show that T881 gets phosphorylated in response to red- and blue light and enhances the light-induced stomatal opening. Moreover, the authors show that T881 is not only involved in regulation of H⁺-ATPases in guard cells, but also in mesophyll cells and hypocotyls in response to auxin. I regard it very likely that this study will attract the attention of many researchers in the field and stimulate further research.

The data presented in the manuscript are in general clear and consistent, with the exception of stomatal conductance, which should be presented as absolute values, as explained in the major points of concern. Unfortunately, the result- and discussion sections are not easy to understand and fail to explain the logic of the experimental approach. To my opinion these sections should be reworked in such a way that also non-expert scientists can read and understand the work.

Major points of concern:

1. The stomatal conductance data in Fig. 2d are given as “relative value to gsw”, but it is not explained how these values were calculated and why the data are presented in this way. It is common to give stomatal conductance as absolute values of the water conductance, relative to the leaf surface. This would enable the reader to see, if the conductance of the plant lines already differs at the start of the experiment. Note that plants with wide open stomata are likely to show a smaller increase in stomatal conductance in response to light, as compared to the control.

Response:

We are sorry for lacking the detailed explanation in the figure legend. The relative value of gsw was calculated by subtracting gsw at 0 min. As shown in Unpublished Fig. 1, the deviations are large, since absolute value of gsw tends to vary in each experiment. Therefore, we often evaluated stomatal conductance as differences (e.g. Hayashi et al., Sci. Rep. 2017). Thus, we would like to leave as it is. However, “relative value of gsw” is not appropriate, we changed the vertical title of stomatal conductance to “Changes in gsw” and added the description about calculation of the value in the Figure legends.

Unpublished Fig. 1 Absolute value of gsw in light-dependent changes of stomatal conductance.

Leaves from dark-acclimated plants were illuminated with red light (Red: 600 $\mu\text{mol m}^{-2} \text{s}^{-1}$) and/or blue light (60 $\mu\text{mol m}^{-2} \text{s}^{-1}$) as indicated. Data are mean \pm SD of four independent experiments.

Moreover, the proper control in Fig. 2d would be the *aha1-9* mutant, complemented with the non-modified AHA1 gene. Please include the stomatal conductance response of this line in this graph.

Response:

Thank you for pointing this out. We have confirmed that WT AHA1 complement transgenic plants showed identical stomatal conductance compared to Col-0. Therefore, we showed Col-0 as a control in Fig. 3d (Previous Fig. 2d). To show the conductance of complement lines, we added the results in Supplementary Fig. 3c and d.

2. In the discussion (line 275) it is mentioned that genomic data for *Vicia faba* are missing. This comes as a surprise to me, as the genome of broad bean has been sequenced and the data were published in Jayakodi et al., 2023 (Nature <https://doi.org/10.1038/s41586-023-05791-5>).

Response:

Thank you for your important comment. When we submitted the first manuscript in February 2023, the genomic data was not opened. Therefore, we modified the sentence regarding missing of genomic information in *Vicia faba*, as follows.

Page 6, line 116-118.

However, because genomic information for *Vicia faba* had been lacking at start of this study, its proteomic analysis in this work was achieved by constructing an expression database from the plant.

In addition, we cited Jayakodi et al. Nature 2023, as follows.

Page 6, Line 118-119

Note that, later, the genome of *Vicia faba* has been sequenced.³⁷

3. The bars in the graph of Figure S5d seem to be colored the wrong way around. The shorter bars should probably be red and the longer bars blue.

Response:

We are very sorry for our mistakes. We corrected the color of the bars in Figure S4d (Previous Figure S5d).

Below I will point out a number of points that need attention, but I think that in general the authors need to rework most of the text in the results and discussion.

1. In the introduction, one can get the impression that the authors did not show that the phosphorylation of T881 is also enhanced by red- and blue light in *Arabidopsis* guard cells. Please modify the text, so the reader understands that light-induced phosphorylation of T881 in guard cells is conserved for broad bean and *Arabidopsis*.

Response:

Thank you for your valuable suggestion. We added the description about the red- and blue light induced phosphorylation of T881 in *Arabidopsis* guard cells as follows.

Page 5, line 92;

Furthermore, red- and blue light-induced phosphorylation of Thr881 were also observed in guard cells from *Arabidopsis thaliana*.

2. Results lines 100-112. The first sentences of the results are a repetition of information that is already given in the introduction. It would make more sense that the authors explain that they performed a phosphoproteomics analysis with guard cells of *V. faba*, because of the

high quality and quantity of phosphoproteins that they can obtain with this species. In the manuscript this information is given at the start of the discussion, but it would make more sense to explain the choice for *V. faba* at the start of the results.

Response:

Thank you for your important suggestion. Due to the connection with other contexts, comprehensive revisions were not possible; however, to the extent possible, I have revised the relevant sections of the Results and Discussion in accordance with your comments. I would appreciate your understanding.

3. Results lines 113-124. The impact of red- and blue light on phosphorylation on the positions T881 and T948 can be explained more clearly. As far as I understand, both red- and blue light promote phosphorylation of T881, whereas the phosphorylation at T942 is only induced by blue light in isolated epidermis and guard cells protoplasts. This could be explained more clearly.

Response:

Thank you for your suggestion. We modified description.

4. Results lines 126-139. How did the authors conduct the experiments with *Arabidopsis* stomata. Did they use epidermal fragments, isolated epidermal peels, or intact leaves. This is not well explained and because of this, the reader does not get a good feeling of the conditions used for these experiments.

Response:

Thank you for your suggestion. We modified description.

5. Results lines 148-152. Why are the results with the *bhp-1* mutant shown in the supplementary figures? The impact of loss of BHP on the phosphorylation status of T881 is unexpected and show that BHP is not required to phosphorylate this position of the H⁺-ATPases.

Response:

Thank you for pointing this out. We moved the results with *bhp-1* mutant to Figure 2. In addition, as we mentioned in the response to #10 comment, we further discussed about the result observed in *bhp-1* GCPs in the Discussion.

6. Results lines 153-156. Here phosphoprotein data obtained with *V. faba* guard cells are mentioned, which are not related to the previous text. These data could be better be moved to the start of the results in which the *V. faba* phosphoproteome data are discussed.

Response:

Thank you for your suggestion. We moved these sentences to the front of Results.

7. Results line 166. Here it is written that transgenic plants carrying wild type AHA1 in *aha1-9* show the same light-induced stomatal opening as wild type (Figure S4). Does this mean that the experiment in Figure S4 was carried out with intact plants?

Response:

In Supplementary Fig. 3 (Previous Supplementary Fig. 4), we performed stomatal aperture measurement using isolated epidermis. We could not find difference between Col-0 and WT AHA1 complement lines. In addition, as described in response to Major points of concern #1, we added the result of stomatal conductance measurement in Supplementary Fig 3c and d.

8. Results lines 193-217. In the first sentence of the section, it is explained that red light causes phosphorylation of the T948 in intact leaves, but the remaining of the text does not fit to this. The authors should explain why they tested the impact of light on the phosphorylation of AHA proteins in mesophyll, and the phytohormone auxin in hypocotyl cells.

Response:

Thank you for your suggestion. We revised and added some sentences why we tested using mesophyll cells and hypocotyls in Results.

9. Results lines 234-250. Please give a reference that explains the nomenclature of the PP2C proteins.

Response:

According to your suggestion, we added accession numbers of genes investigated in this study including PP2Cs with reference in the Methods section (Page 25, Line 727-735).

10. Discussion, lines 288-293. Please discuss the unexpected results with the bhp-1 mutant in further detail. Does the phosphorylation of T881 fit to the phenotype of this mutant?

Response:

Thank you for pointing this out. We added descriptions about Fig. 2 in Discussion. Regarding your suggestion "Does the phosphorylation of T881 fit to the phenotype of this mutant?", it is very important question. Actually, higher phosphorylation level of Thr881 does not fit with the closed stomata phenotype in bhp, because Thr948 phosphorylation defects in bhp mutant. These results indicate that Thr948 phosphorylation is more effective for its function than Thr881. However, these explanations are very complicate. We think that further supportive experiments are needed. Therefore, we don't mention about this point in this manuscript.

11. Discussion, lines 336-338. Here it is written that "T881 phosphorylation of the PM-H⁺-ATPase contributes around 50% to ...". How did the authors calculate this number and to which conditions do they refer?

Response:

We estimated this value according to the results of light-induced stomatal opening in Fig 3c (Previous Fig. 2c). We cited Figure in the sentence.

12. Figure 5. It should be mentioned in the legend that this scheme reflects the situation of isolated stomata, as or intact leaves red light also promotes phosphorylation of T948.

Response:

Thank you for your important suggestion. In the previous manuscript, we illustrated the scheme which reflects the situation of isolated stomata. In the present manuscript, inspired by your suggestion, we have modified the scheme that includes mesophyll cells to explain responses closely related to the intact leaf situation in Fig. 6 (Previous Fig. 5) and added the sentences explaining new scheme in Discussion.

Reviewer #4 (Remarks to the Author):

The manuscript entitled "Phosphorylation of plasma membrane H⁺-ATPase Thr881 participates in light-induced stomatal opening", describes the AHA1 phosphorylation of Thr881 and its role in PM H⁺-ATPase function in stomatal guard cells in response to red and blue light. The authors conclude that Thr881 phosphorylation by both red and blue light influences plasma membrane AHA1 function in stomatal guard cells and contributes to stomatal opening. The results suggest that this phosphorylation event is regulated by phototropin-mediated blue light-signaling and guard-cell photosynthesis. This manuscript has interesting results but falls short in demonstrating their results and several conclusions

are too speculative. Also, I fully support the assessments made by the two referees, particularly with regard to two crucial points.

Firstly, the decision to perform a phosphoproteomic screen in *Vicia faba* while restricting the functional validation of AHA1 (vs AHA2) exclusively to *Arabidopsis* raises notable concerns. Although the authors' attempt to clarify the experimental design is commendable, the interpretation seems somewhat strained, since there are inherent differences between these plant species. For example, FC treatment did not induce the same effects on T881 phosphorylation in *Arabidopsis thaliana* and *Vicia faba* (Fig. 1 and Supplementary Table 3). This deviation from a proper *in vivo* analysis raises questions about the rationale for this approach.

Response:

Thank you for your thoughtful comments. Actually, at the start of the experiments, phosphorylation of Thr881 could not be detected in *Arabidopsis* GCPs due to insufficient information. In contrast, by utilizing *Vicia faba*, which allow obtaining a large quantity of high-purity GCPs, we successfully identified the phosphorylation of Thr881 and proceeded with the study. As you pointed out, there may be slight variations, such as in responsiveness to FC. In this study, we concluded that the important role of Thr881 phosphorylation for PM H⁺-ATPase function in guard cells based on the analysis using transgenic plants of *Arabidopsis thaliana*. Therefore, we have decided to include "in *Arabidopsis thaliana*" in the summary sentence which describes importance of Thr881 phosphorylation, of Abstract, Introduction, Results and Discussion sections.

Secondly, I fully agree with the observation that the manuscript tends to be highly descriptive. While obtaining a panoramic view of the stimuli correlated with phosphorylation of T881 and T948 is valuable, the absence of a more in-depth mechanistic investigation may reduce the overall impact of the manuscript. In particular, the authors discuss potential candidates for protein kinases involved in the phosphorylation of Thr881 and raise questions about the mechanism by which red light induces this phosphorylation. Striking a balance between presenting an overview and delving deeper into mechanisms could improve the comprehensiveness of the manuscript and make it more appealing to a wider audience. Consider incorporating more in-depth analyses to strengthen the scientific basis of the manuscript and increase its relevance.

Response:

Thank you so much for your important comments. We greatly appreciate the significance of your feedback. According to your suggestion, we revised throughout Discussion and deleted the descriptions regarding Thr881 Kinase Prediction in Discussion and Supplementary Table 4. We hope this improves the manuscript. Moving forward, we would like to make efforts to advance a more in-depth mechanistic investigation and strengthen the scientific basis.